# Reduced adhesion of aged intestinal stem cells contributes to an accelerated clonal drift

Ali Hageb[1], Torsten Thalheim[2], Kalpana J Nattamai[3] , Bettina Möhrle[1], Mehmet Saçma[1] , Vadim Sakk[1] , Lars Thielecke[4], Kerstin Cornils[5,6], Carolin Grandy[7] , Fabian Port[7] , Kay-E Gottschalk[7], Jan-Philipp Mallm[8] , Ingmar Glauche[4] , Jörg Galle[2], Medhanie A Mulaw[9,]* , Hartmut Geiger[1,]* 

Upon aging, the function of the intestinal epithelium declines with a concomitant increase in aging-related diseases. ISCs play an important role in this process. It is known that ISC clonal dynamics follow a neutral drift model. However, it is not clear whether the drift model is still valid in aged ISCs. Tracking of clonal dynamics by clonal tracing revealed that aged crypts drift into monoclonality substantially faster than young ones. However, ISC tracing experiments, in vivo and ex vivo, implied a similar clonal expansion ability of both young and aged ISCs. Single-cell RNA sequencing for 1,920 high Lgr5 ISCs from young and aged mice revealed increased heterogeneity among subgroups of aged ISCs. Genes associated with cell adhesion were down-regulated in aged ISCs. ISCs of aged mice indeed show weaker adhesion to the matrix. Simulations applying a single cell–based model of the small intestinal crypt demonstrated an accelerated clonal drift at reduced adhesion strength, implying a central role for reduced adhesion for affecting clonal dynamics upon aging.

## Introduction

A hallmark of the intestine is cup-shaped crypts harboring intestinal stem cells (ISCs). ISCs are located at the base of the intestinal crypt, flanked by differentiated Paneth cells, followed by a highly proliferating transient amplifying (TA) progenitor cell zone. TA cells further proliferate and differentiate into enterocytes, goblet cells, and enteroendocrine cells that migrate out of the crypt to form the finger-like villi. ISCs replace the intestinal epithelium every 3–5 d to maintain intestinal tissue homeostasis and to supply the intestinal epithelium with TA cells (Barker, 2014). ISCs express the marker protein Lgr5 (Barker et al, 2007; Sato et al, 2009). There are between 12 and 16 Lgr5 expressing cells per crypt. However, only 5–7 out these might be functionally active ISCs (Kozar et al, 2013). Intestinal homeostasis in young animals goes along with a clonal competition among ISCs that can be described by a model of neutral drift or neutral competition (Lopez-Garcia et al, 2010; Snippert et al, 2010; Grun et al, 2015). In this model, individual ISCs in a crypt are almost equipotent but stochastically and continuously acquire the ability to dominate the generation of differentiated cells to the villus for a defined amount of time, while being replaced by clonal expansion of other ISCs that will subsequently dominate contribution to the villus and so on (Snippert et al, 2010). Changes of the competition dynamics have been investigated in mice (Snippert et al, 2014; Huels et al, 2018) and by applying single cell–based computational models (Thalheim et al, 2016). In young mice, reports support that in 7–8 wk, between 50% and up to 75–80% of all crypts turn monoclonal, whereas it might take up to 30 wk to turn all crypts monoclonal (Winton & Ponder, 1990; Li et al, 1994; Lopez-Garcia et al, 2010; Snippert et al, 2010). During the next couple of weeks, one novel, again almost equipotent neutral ISC subclone within the currently monoclonal crypt, will replace the current dominant ISC clone to turn the crypt again monoclonal.

Aging results in a decline of the function of ISCs. This decline includes reduced self-renewal potential and the ability to maintain tissue homeostasis upon stress (Martin et al, 1998; Potten et al, 2001). For example, crypts or ISCs from aged mice present with a reduced percentage of cells that are able to form fully structured organoids in ex vivo assays (Nalapareddy et al, 2017, 2021; Mihaylova et al, 2018; Pentinmikko et al, 2019). Aging of ISCs is thought to be a main driver of aging-related diseases of the digestive track, like malabsorption or cancer (Saffrey, 2014; Jasper, 2020). ISCs are, over the lifespan of the organism, exposed to

[1]Institute of Molecular Medicine, Ulm University, Ulm, Germany  [2]Interdisciplinary Centre for Bioinformatics, University Leipzig, Leipzig, Germany  [3]Division of Experimental Hematology and Cancer Biology, Cincinnati Children's Hospital Medical Center (CCHMC), Cincinnati, OH, USA  [4]Institute for Medical Informatics and Biometry, Technische Universität Dresden, Dresden, Germany  [5]Clinic of Pediatric Hematology and Oncology, Division of Pediatric Stem Cell Transplantation and Immunology, University Medical Center Hamburg-Eppendorf, Hamburg, Germany  [6]Research Institute Children's Cancer Center Hamburg, Hamburg, Germany  [7]Institute for Experimental Physics, Ulm University, Ulm, Germany  [8]Division of Chromatin Networks, German Cancer Research Center (DKFZ), Heidelberg, Germany  [9]Central Unit Single Cell Sequencing, Medical Faculty, Ulm University, Ulm, Germany

Correspondence: hartmut.geiger@uni-ulm.de; Medhanie.Mulaw@uni-ulm.de
*Medhanie A Mulaw and Hartmut Geiger contributed equally to this work.

challenging environments like acids, enzymes, nutrition elements, microbial pathogens, and microbiome composition. Such environmental factors might be a driving force in aging of ISCs. In addition, signals from Paneth cells that form a niche for ISCs have been reported to play a role in diminishing the stemness potential of aged ISCs. For example, Notum, a Wnt antagonist that is primarily produced by Paneth cells, attenuates ISC-driven regeneration of aged intestinal epithelium (Pentinmikko et al, 2019).

Although there is a growing understanding of the molecular mechanisms that underlie aging of ISCs, a more detailed understanding on how aging affects ISC potential and clonal dynamics in vivo is missing. For example, a current paradigm in the hematopoietic system is that there is an increase in heterogeneity among stem cells upon aging. This increase seems to be linked to a lower number of clones (in this case dominant) that contribute actively to blood cell formation, which has been termed clonal hematopoiesis. It is thought that these changes are causal for unwanted aging-associated diseases of the hematopoietic system (Akunuru & Geiger, 2016). ScRNA-seq analyses of young ISCs showed an initially homogenous pool of ISCs (Grun et al, 2015), which however could be further subdivided into two or three subgroups by single-cell expression profiling, based on the intestinal region ISCs were derived from but more importantly, according to their ability to interact with distinct signals from the environment (Haber et al, 2017; Biton et al, 2018). The level of heterogeneity among aged ISCs is not known. Novel knowledge on changes among individual ISCs upon aging will help to further understand consequences of aging of ISCs for tissue homeostasis and likely also for aging-associated clonal outgrowth like in cancer.

## Results

### Accelerated clonal drift in crypts of the aged intestine

Although central to the dynamics of tissue homeostasis and likely also for cancer development, the nature of the ISC clonal drift in aged animals is not known. To assess the clonal drift within crypts of aged animals, we used an established model of long-term lineage tracing. To this end, Tg(Vil-cre/ERT2)23Syr; *Gt(ROSA) 26Sor^{tm1(CAG-Brainbow2.1)Cle}*/J mice were generated (hereafter, vil1-cre; R26 Confetti). Upon one dose of tamoxifen (tam), a large frequency of intestinal cells, including ISCs, of either young (12–16 wk) or old (80–85 wk) mice were labeled by one of the four confetti fluorophores: cell membrane cyan (CFP), cytoplasmic yellow (YFP), cytoplasmic red (RFP), or nuclear green (nGFP) (Figs 1A and S1A and B). Similar to previous publications using this intestinal tracking system in mice (Snippert et al, 2010), the frequency of recombination for nuclear GFP was lower than 1%, and thus, analyses of nGFP were not further pursued in our analyses. Quantification of the frequency of the three remaining fluorophores in young and aged crypts 5 d post tam induction confirmed almost random and similar labeling efficiencies among young and aged crypts (Fig S1B), which excludes a possible bias in recombination efficiency in young versus aged intestinal cells as a confounding factor for our investigations. Intestinal crypts in animals were subsequently analyzed by whole-mount confocal microscopy (Fig 1A and B).

5 d post–tam induction, crypts of both young and aged animals showed a random mixture of the confetti fluorophores (Fig 1B and E and Videos 1 and 2). However, already 4 wk post-induction, we observed an elevated percentage of crypts in aged animals that were biclonal or monoclonal compared with crypts in young animals (Fig 1B–D). Between week 4 and 8 post-tam, the frequency of biclonal crypts in aged mice declined rapidly with a concomitant increase in the frequency of monoclonal crypts, although many of the young crypts still showed contribution of at least two to three clones (Fig 1B–D). 20 wk after initiation of labeling, almost all aged crypts were monoclonal, whereas a considerable proportion of young crypts still remained biclonal (Fig 1B–D). We also determined whether crypts with the same color were stochastically distributed or distributed as clusters, which would indicate that drift might affect multiple crypts (Fig S1C). Median absolute deviation analyses revealed that the distribution of monoclonal crypts was indeed nonrandom in both young and aged crypts, whereas the level of nonrandom distribution of monoclonal crypts with the same color was significantly elevated in aged animals (Fig S1D). This implies that upon aging, there might be an accelerated spreading of dominant clones among neighboring crypts, which could be linked to the accelerated clonal drift but also be due to enhanced crypt fission and fusion upon aging (Bruens et al, 2017). In aggregation, our data show that upon aging, crypts still show a clonal drift and clonal succession but with a very much accelerated turnover of the dominant clone compared with the turnover in young crypts. In addition, clonality is also more likely to spread among aged crypts compared with young crypts. Whether both of these findings are mechanistically linked will need to be further investigated.

### Aged ISCs do not show accelerated expansion

Clonal drifts in young animals are driven by ISCs (Snippert et al, 2010; Sei et al, 2019). The accelerated drift upon aging might thus be linked to an accelerated expansion of ISCs and their offspring upon aging. We therefore determined the level of expansion of Lgr5+ ISCs and their offspring over a 5-day time period. For this purpose, B6.129P2-Lgr5^{tm1(cre/ERT2)Cle}/J; *Gt(ROSA)26Sor^{tm1(CAG-Brainbow2.1)Cle}*/J mice were generated (hereafter Lgr5e-creERT2; R26 Confetti, Fig S2A). ISCs from such mice, upon tam activation, express besides the eGFP reporter linked to the expression of Lgr5 also confetti colors only in ISCs (Lgr5+ cells), with a color frequency distribution similar to the one reported in Fig S1B. Flow cytometric analyses indicated a similar frequency of Lgr5 ISCs–high and –low cells in intestine of young and aged animals (Fig S2B and C). Consistently, also the number of Lgr5eGFP cells within individual crypts was similar among young and aged mice (Fig S2D). In addition, the vertical extension of the Lgr5eGFP region (Fig S2E and Videos 3 and 4) was similar between young and aged crypts, which excluded all these parameters as confounding factor in our following analyses.

To visualize the level of expansion of individual Lgr5 ISCs within individual crypts, fluorescent confetti colors in Lgr5+ cells of young (12–16 wk) or aged (80–85 wk) mice were initiated by injection of a single dose of tam (Fig 2A). The number of crypts with at least one labeled Lgr5 cell was between 5% and 10% at day 1 post–tamoxifen injection and similar between young and aged animals, indicating a

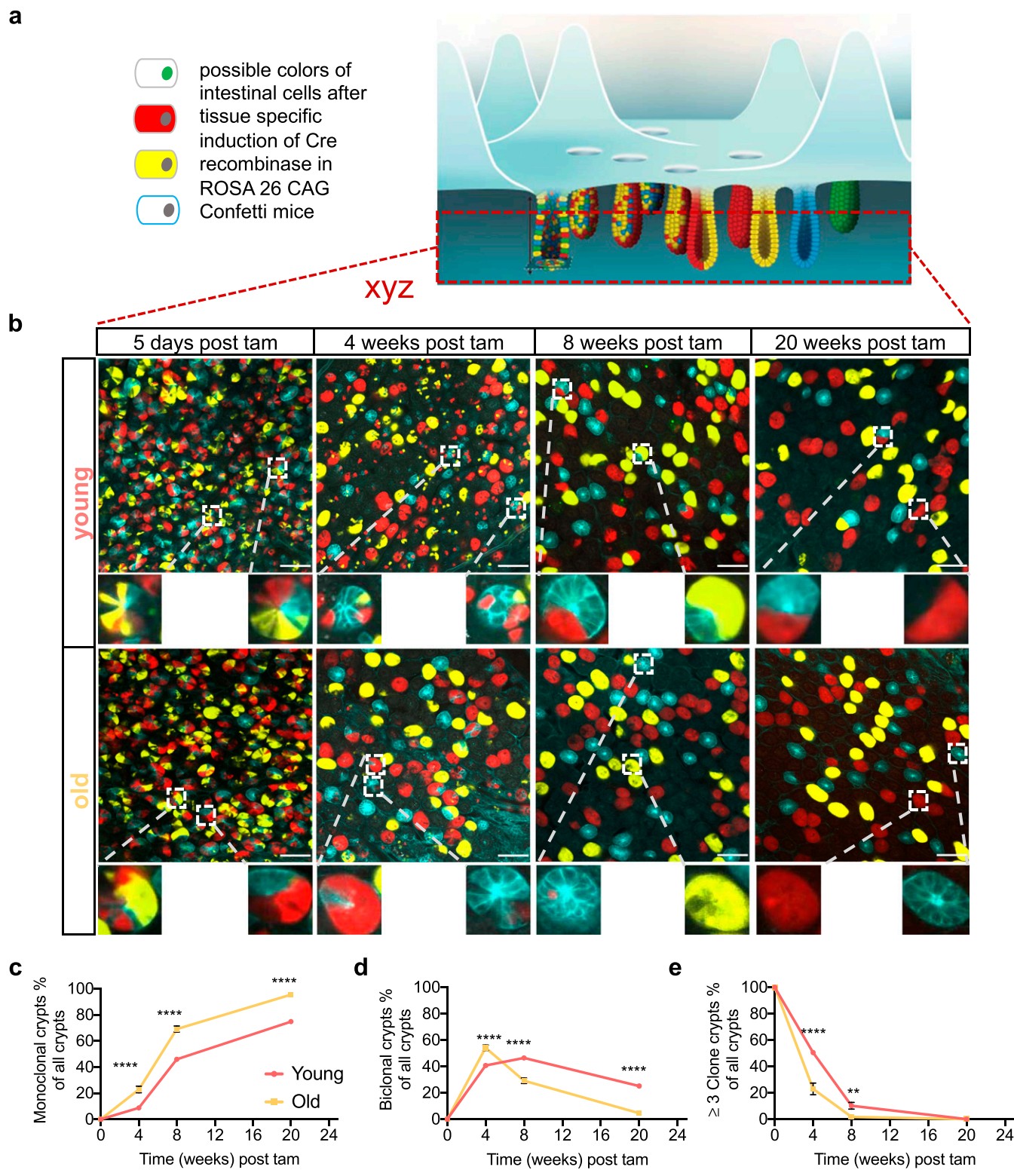

**Figure 1. Accelerated clonal drift in crypts of the aged intestine.**
**(A)** Upon tamoxifen injection, intestinal epithelial cells, including intestinal stem cells in villin-cre Confetti mice will acquire on of the confetti colors. Over time, crypts will become monochromatic. **(B)** Representative images of crypts bases at different time points post tam injection/confetti initiation in whole-mount intestinal tissue. Scale bar = 100 μm. **(C, D, E)** Quantification the frequency of monochromatic, dichromatic, and multi-chromatic crypts in individual young and aged crypts between 5 d and 20 wk post–confetti induction. Analysis have been performed on confocal z-stacks (each stack has 15–22 layers with 3.6 μm distance for each) covering the whole length of all the crypts analyzed. Shown are means ± SEM. Analysis have been performed on confocal z-stacks (each stack has 15–22 layers with 3.6 μm distance for each) covering

similar labeling efficiency of young and aged Lgr5 cells (Fig S2F). At 3 d post-tamoxifen, up to 60% of crypts were labeled in both young and aged animals. At 5 d post-induction, the frequency of labeled crypts started to decrease for both ages groups (Fig S2F).

We subsequently calculated the vertical expansion gained by offspring of individual Lgr5 cells at day 3 and 5, as well as the total area covered in all image stacks. We focused on the YFP label as the YFP signal was simply best suited for semiautomated image analysis. Surprisingly, the relative distribution of the vertical expansion within the crypt at day 3 and day 5 post-induction was almost identical among young and aged crypts (Fig 2B). Similarly, there was no difference in the total YFP positive area among young and aged crypts (Fig 2C). These data imply that aged ISCs do not show accelerated expansion in vivo in comparison to young ones. This finding excludes heterogeneity in proliferation an expansion upon aging as a central mechanism for the accelerated drift upon aging. It implies other mechanism like heterogeneity in the competitive potential of ISCs upon aging as contributors to the accelerated drift.

### Organoids from aged crypts show a clonal complexity similar to young organoids

Both a clonal drift driven by a more heterogeneous pool of ISCs and selection of clones with so far only minor selective advantage because of a small number of mutations however remain consistent with the observed accelerated clonal drift. Ex vivo organoids formed from ISCs within crypts are an accepted and valuable model system to study ISCs function ex vivo (Sato et al, 2009; Lukonin et al, 2020; Nikolaev et al, 2020). We used a recently developed, mid-complexity (about 350 barcodes) but fully annotated lentiviral barcode library (Thielecke et al, 2017) to determine changes in clonal expansion of ISCs from young and aged crypts upon ex vivo organoid culture (Figs 3A and S3A). To this end, crypts and thus ISCs in crypts from young or aged animals were transduced with the barcode library and subsequently cultured and expanded under organoid culture conditions. In these experiments, on average, about 50% of organoids were transduced by the barcode virus, with a similar transduction frequency of young and aged crypts (Fig S3B). The overall number of barcode libraries retrieved was similar in both age-groups and showed a high level of representation of complexity (on average 75 barcodes per analysis, Fig S3C). A high level of transduction, an equal level of transduction efficiency and the high level of complexity obtained upon retrieval ensured that the barcode tracing experiments are representative of all crypts in the experiment and comparable between experiments performed with young and aged crypts. Organoids formed from old crypts (O1–O5) showed no difference in the frequency of samples in which a small number of barcodes dominated the relative abundance of barcodes in a given sample (Fig 3B). Consistent with no significant difference in clonality parameters in young and aged organoids, the

relative contribution of the most abundant barcode (Fig 3C), as well as the Shannon diversity index, which is a measurement of barcode diversity (Fig 3D), is not different between young and aged organoids. These data are consistent with no or only a very minor advantage of individual aged ISCs in conferring dominance upon ex vivo expansion over young ISCs.

### Increased transcriptional heterogeneity among aged ISCs

To obtain direct insight into the level of transcriptional heterogeneity among aged ISCs, 1,920 individual ISCs (Lgr5$^{high}$ cells, Fig 4A) from 3 young and 4 aged animals were analyzed by single-cell RNA sequencing (scRNA-seq) using a SMARTseq2 approach to provide a high level of sequencing depth and transcript coverage. We first investigated, via a deep learning algorithm, whether there is a gene expression signature that was able to discriminate between young and aged ISCs. The training of the deep learning model was performed using 65% of the cells, whereas the remaining 35% were used for validation of the prediction accuracy of the model. After training, the algorithm achieved a prediction accuracy of at least 96% on the unseen data (Fig S4A), demonstrating that the transcription profile of individual young ISCs is predominantly distinct from the profile of individual aged ISCs. In the next step, we generated a correlation coefficient matrix for all the expressed genes and measured their degree of correlation to the prediction model. Principal component analyses were performed using the top 5% of positively (higher aging likelihood predictors) or negatively correlated (higher young likelihood predictors) genes (Table S1). We observed that young and aged ISCs showed clear segregation when plotted using the first 3 principal components (PCs) (Fig 4B). A gene set enrichment analysis (GSEA) using a pre-ranked list of genes based on their correlation coefficient to the prediction model (the aging signature) showed a significant negative correlation to a previously reported ISC signature (on bulk ISCs) from young mice ([Muñoz et al, 2012]; Fig 4C). This indicates that our deep-learning–based aging signature significantly shares genes that are also implicated in stemness of ISCs, implying concordance between aging and stemness axis. We next performed gene ontology biological process (GO-BP) GSEA of the pre-ranked aging signature genes. Biological adhesion, intracellular signal transduction, and tissue morphogenesis were negatively correlated with the pro-aging subset and thus positively correlated with the young like phenotype (Figs 4D and S4C). On the other hand, GO terms including small-molecule metabolic process, oxidation reduction process, small-molecule biosynthetic process, and cofactor metabolic process were positively correlated with the pro-aging signature (Figs 4E and S4B and C). Changes in these pathways might be critical for aging of ISC and might thus also contribute to the accelerated drift.

To determine the level of transcriptional heterogeneity among individual ISCs within individual animals, graph-based nearest

---

the whole length of all the crypts analyzed. N = 3–5 mice (young or old for each time point analyzed). 3–5 stacks per animal and time point and on average of ~80 crypts for each stack were analyzed. The data from all z-stacks from one animal were averaged to obtain a single value on clonality per mouse per time point. In total, 3,437 (1,890 young and 1,547 old), 2,627 (1,188 young and 1,439 old),1,644 (716 young and 928 old), and 1,956 (780 young and 1,176 old) crypts were analyzed for the 5 d, 4, 8, and 20 wk time points. Shown are means with SEM, ** = P < 0.01, *** = P < 0.001, **** = P < 0.0001, two-way ANOVA.

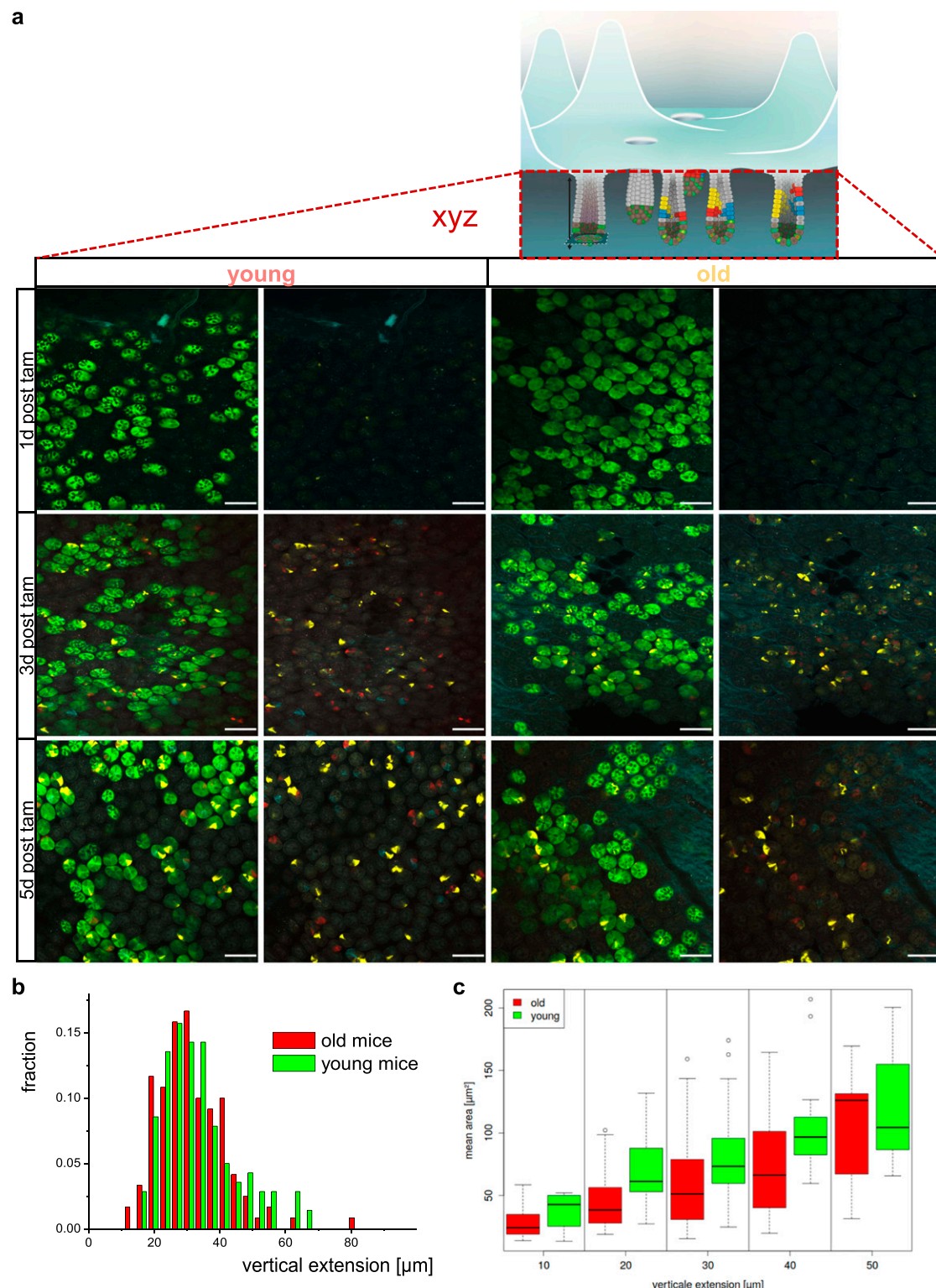

**Figure 2.  Expansion of Lgr5-positive intestinal stem cells is not altered by aging.**
**(A)** Representative images of crypt bases at different time points post–tam injection/confetti initiation in whole-mount intestinal tissue of young or old Lgr5CreConfetti mice. In these animals, Lgr5+ positive cells show cytoplasmic green in addition to the confetti colors. Scale bar = 100 μm. **(B)** Distribution of the vertical extension of clones in young and aged crypts 5 d post–tam induction as seen by the YFP signal. **(C)** Average area covered by the YFP signal of clones in young and aged crypts 5 d post–tam induction. Clones within defined ranges of vertical extension have been clustered (two young and two old mice, between 60 and 90 clones per mouse). Few clones with an extension above 55 μm were omitted.

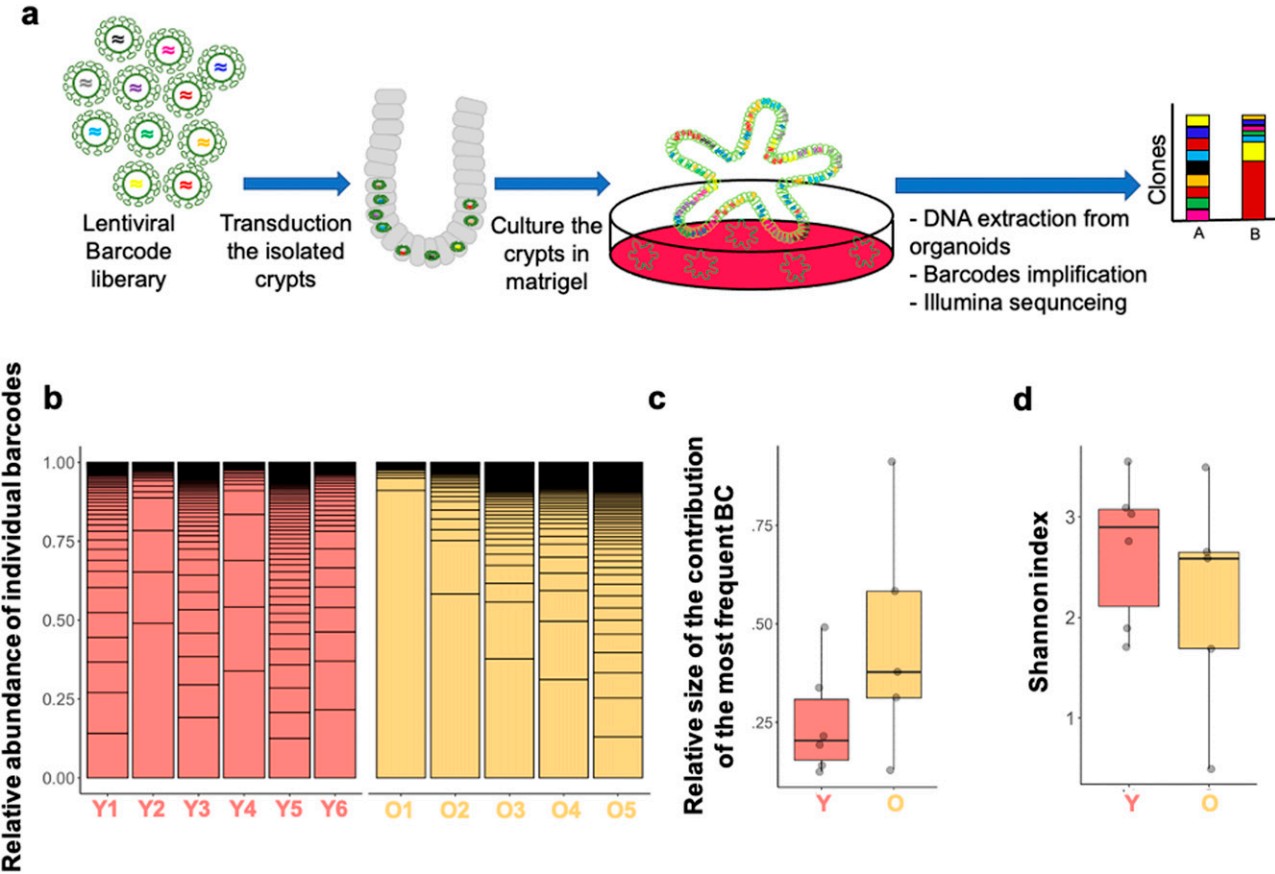

**Figure 3. Aged intestinal stem cells show only minimally reduced clonality ex vivo.**
**(A)** Experimental setup: crypts from either young (12–16 wk) or aged (≥80 wk) mice were transduced with a lentiviral-based barcode library, and DNA extracted from organoids stemming from the crypts 7 d post-transduction was subjected to sequencing to quantify barcodes. **(B)** Relative abundance of the most represented barcodes among organoids from young and aged crypts (six young and five aged individual mice). **(C)** Relative size of largest barcode clone among young or old organoids. Mann–Whitney U-test: $P = 0.25$. **(D)** Shannon index (diversity) of barcodes retrieved from organoids from young or aged crypts. Mann–Whitney U-test: $P = 0.33$.

neighbor clustering analyses for all ISCs from each individual mouse were performed. Although the number of optimal gene expression-based clusters of ISCs remained similar among young and aged animals (on average 4, Figs 4F and S4D), the median Euclidean distance among clusters was higher among ISC clusters in old animals compared with young animals (Fig 4G). As the analysis also took into account within cluster distribution, in aggregation, the results showed that aged ISCs formed more compact and distinct clusters comprising cells with high transcriptional similarity within a cluster but more distinct from cells in other clusters.

An unsupervised hierarchical clustering-based heatmap of the individual clusters, using the top 5% aging predictors genes, further showed segregation of gene expression profiles of cells by cluster membership (Fig 5A), implying that clusters are structured/ordered in concordance with the top aging signature genes. In the next step, we performed pairwise GSEA between clusters from individual animals, to assess which of the clusters show significant correlation to the top 5% gene signatures to order the clusters according to the correlation to aged and young signatures. Subsequently, we performed diffusion map analyses. A pseudo-time order of the clusters from the GSEA are indicated by arrows showing

directionality of the order of the clusters toward an aged expression signature (Figs 5B and S4E). We noted that young samples show a nondistinct order of the clusters and thus no strong pseudo-time relationships, whereas in the aged individuals, we observed that the clusters segregated along an aging profile pseudo-time axis. This implies that, among aged ISCs, cell clusters are driven by step-wise transition to aging along the ISC aging signature by step-wise acquiring the aging transcription program. The accelerated clonal drift upon aging thus correlates with an increase in heterogeneity among groups of ISCs from old animals.

## Reduced adhesion of aged ISCs is caused by reduced Wnt signaling

Adhesion-related processes were among the differentially expressed Gene Ontology terms in the GSEA analysis based on a correlation of the GO signatures to the ISC aging signature (Fig 4D). A more in-depth analysis revealed that indeed adhesion related GOs/genes were primarily down-regulated in aged ISCs (Fig 6A). Changes in adhesion of ISCs have been linked to change in clonal dynamics in cancer (Dalerba et al, 2007). To experimentally determine whether aging alters adhesion of aged ISCs, a shear stress

**Figure 4. Intestinal stem cell (ISC) single cell aging gene expression signature reveals increased heterogeneity among clusters of aged ISCs.**
**(A, B, C)** Schematic overview of Lgr5-high ISCs sorting for scRNA-seq. (B) Principal component analysis classification of young and aged ISCs depending on the 5% predicted genes. (C) Gene set enrichment analysis of ISCs gene signature upon aging. **(D, E)** Gene Ontology reactome pathways network of networks increased (blue) or decreased. (red) in young ISCs. **(F, G)** Clusters of similar expression among young or aged ISCs (examples). (G) Euclidean distance between clusters of ISCs.

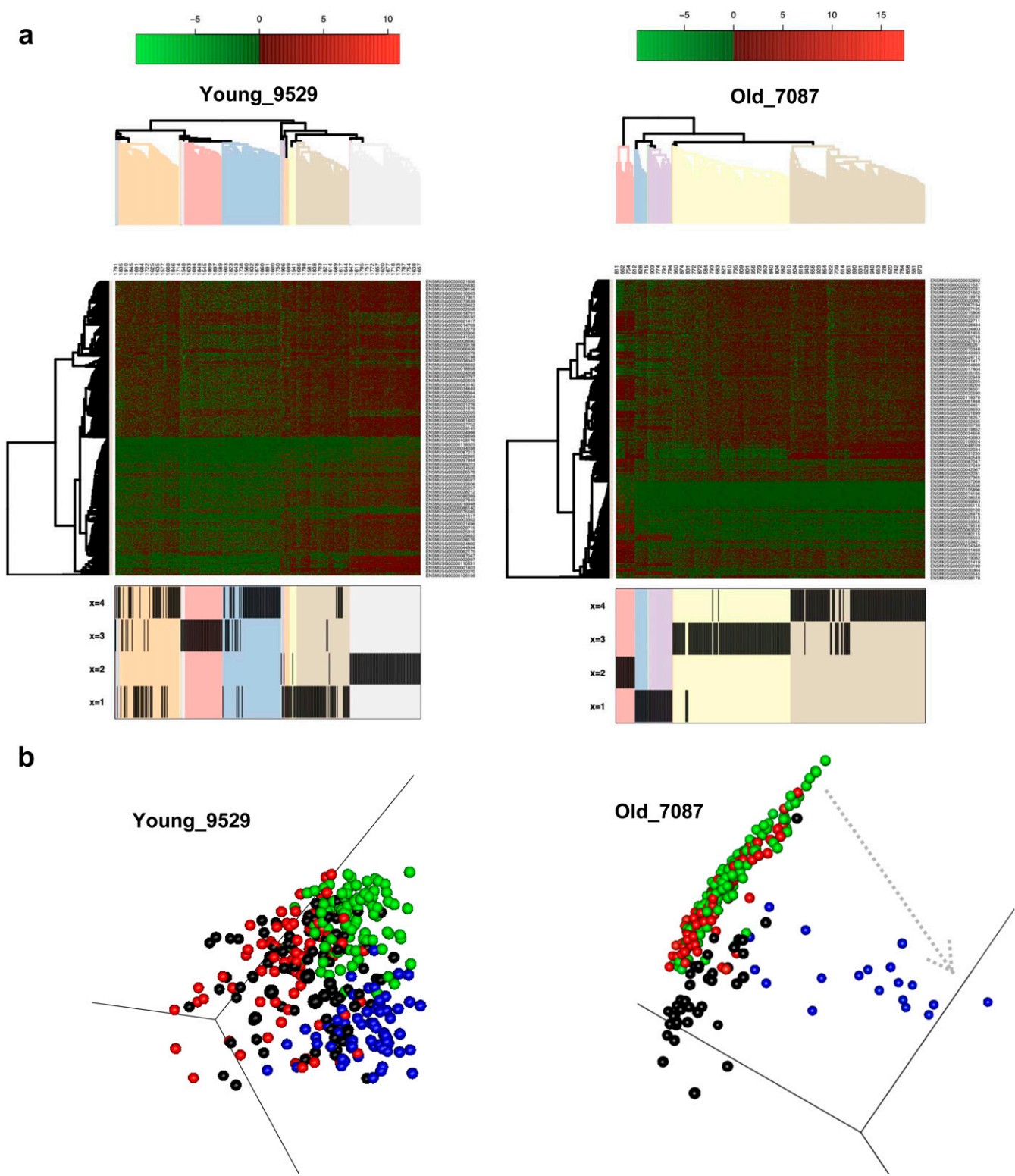

**Figure 5. Clusters from aged intestinal stem cells segregate along an aging profile pseudo-time axis.**
**(A)** Gene expression heatmap for gene clusters defined in Fig 4F. The color scale shows level of expression, and the hierarchical clustering arranges cells according to similarities in expression levels of these aging signatures identified (Fig 4A–C). The lower part of the figure identifies the intestinal stem cell cluster of that mouse (see Figs 4F and S4D) the individual cell has been assigned to. These are, for example, four clusters in young mouse #9529 or old mouse #7087. **(B)** Diffusion maps of the clusters with pseudo-time arrows (examples).

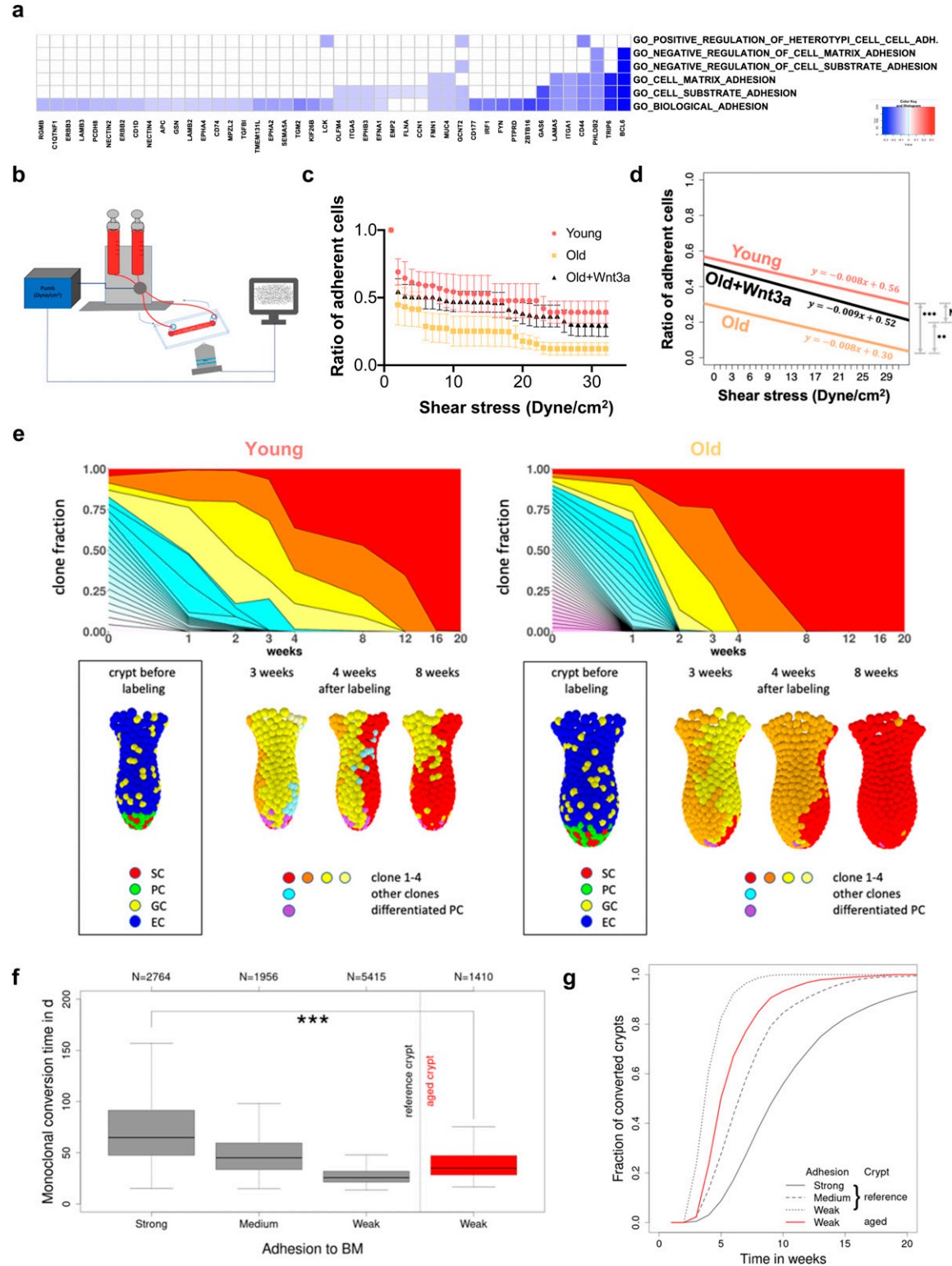

**Figure 6. The accelerated clonal drift upon aging is linked to reduced adhesion of intestinal stem cells (ISCs) caused by reduced Wnt signaling.**
**(A)** Heatmap of the level of expression of adhesion genes (x-axis) in GO terms linked to adhesion (y-axis, right) in aged ISCs. All listed genes are down-regulated in aged HSCs. **(B)** Experimental setup for determination of adhesion to Matrigel in response to shear stress. **(C)** The ratio of adherent ISCs on a collagen IV/Matrigel–coated chamber upon a continuously increased level of shear stress. Shown are means with SEM. Mice, n = 3 young, 3 old, and 3 old Wnt3a-treated group. **(C, D)** Nonparameteric regression analysis of the ratio of adherent cells depicted in (C). As there were no significant differences between the slopes, the intercepts were compared using a nonparametric test of proportions using the chi-square goodness of fit test. ** = $P < 0.01$, *** = $P < 0.001$. **(E)** Examples for a crypt from a young (left) and an old mice (right).

model was applied to quantify the number of Lgr5 ISCs that remain adherent to Matrigel in response to a continuous, linear change in external force (Fig 6B). Overall, young ISCs resisted detachment by shear stress significantly better than aged ISCs, especially at a force higher than 20 dyne/cm$^2$ (Fig 6C and D). Aged ISCs are therefore impaired in their ability to adhere to matrix. Aged ISCs show reduced canonical Wnt signaling (Fig S4F and Nalapareddy et al [2017]), which drives aging of ISCs, whereas addition of Wnt3a to aged ISCs restores canonical Wnt signaling and, at least in part, rejuvenates ISC function (Nalapareddy et al, 2017; Pentinmikko et al, 2019). Mechanistically, reduced Lgr5 can result in a loss of cortical F–actin accompanied by weaker cell–cell interactions (Carmon et al, 2017). Together, these data suggest that reduced Wnt-signaling in aged ISCs might affect their adhesion properties. Indeed, aged ISCs, when exposed to Wnt3a, showed significantly improved adherence, almost to the level seen in young ISCs (Fig 6C and D). These data demonstrate that reduced Wnt signaling in aged ISCs is sufficient to cause reduced adhesion of aged ISCs.

### Reduced adhesion accelerates clonal drift in a computational model of the crypt

Genetic or pharmacological modifications of adhesion in vivo are usually nonspecific (both with respect to pathway and cell type) and nongraded. Consequently, it remains very difficult to experimentally investigate the consequences of graded levels of in vivo adhesion for the regulation of the dynamics of the clonal drift. We therefore used an individual cell–based computational model of the intestinal crypt to determine the influence of reduced adhesion properties on competition of individual ISCs in crypts over time (Fig 6E). The model has been already successfully applied to simulate long-term tissue dynamics under homeostasis and during transformation (Buske et al, 2011; Thalheim et al, 2016). In the extended model used for these analyses, the matrix was represented by a dense polymer network, and we introduced an additional adhesion structure of niche cells that attaches these cells to individual matrix polymers.

A 10% reduction of the adhesive force at which the linkage to individual matrix polymers breaks is sufficient to result in shortening of the conversion time from about 10 to 4–6 wk, similar to what has been observed in aged mice (Fig 6F and G). Considering differences in the crypt shape between young and aged mice (Fig S5A), in agreement with our experimental observations (Nalapareddy et al, 2017), we were able to show a similar short-term clone size distribution as actually found in mice (Fig S5B). Thereby, an increased crypt diameter ensures constant ISC numbers during aging, which otherwise decrease with reduced adhesion strength. The model thus correctly mirrors our observations in aged animals (Figs 1 and 2). We further determined (Fig S5C) the expression

pattern of the GO adhesion genes (Fig 6A) in all ISCs from each individual mouse and compared it to their cluster membership (Figs 4F and S4D). The color scale in Fig S5C shows expression, and the hierarchical clustering arranges cells according to similarities in expression levels. The lower part of the figure identifies the cluster the cell has been associated to (young or old mice). There is no obvious pattern in the association of the genes with distinct clusters, so cells in individual clusters do not show a distinct expression pattern of adhesion genes, that is, clustering is not driven by changes in adhesion molecules but rather by differences in the expression of genes linked to overall stem cell aging (Fig S4E). In summary, the computational model explains accelerated monoclonal conversion in aged mice by reduced ISC adhesion without assuming accelerated cell proliferation. Our results provide clear evidence that accelerated clonal drift upon aging is tightly linked to reduced adhesion of aged ISCs caused by reduced canonical Wnt signaling in ISCs and thus uncoupled from division.

## Discussion

The clonal dynamics of ISCs upon aging is not well understood, although the current paradigm holds that changes in such dynamics contribute to aging-associated intestinal diseases such as cancer. We report here an accelerated clonal drift within intestinal crypts of aged animals. This however is not based on differential expansion of young in comparison to aged ISCs but rather linked to reduced adhesion of aged ISCs because of reduced canonical Wnt signaling. Our data on the dynamics of the drift in young animals are similar to the dynamics described by Lopez-Garcia (Lopez-Garcia et al, 2010) which also report close to 50% of all crypts being monoclonal 8 wk after induction. On the other hand, in our experiment transition to monoclonality is somewhat delayed when compared with data reported by Snippert et al (2010), which could be because of differences in tissue analysis (whole-mount staining versus thin-sections) or because of differences in environmental factors. This accelerated drift upon aging correlates with an increase in heterogeneity among aged ISCs. Previous studies reported that Wnt signaling reduction accelerated the dominance of mutated cells in the crypt base (Yan et al, 2017; Huels et al, 2018) and that in general, Wnt signaling plays an important role in the adhesion (Amin & Vincan, 2012). For example, ISCs localized in distal border of the niche rapidly lose stemness and tended toward differentiation (Ritsma et al, 2014). Furthermore, it was reported that adhesion mechanism showed an important role in the competition between somatic cyst stem cells (CySCs) and spermatogonial stem cells (SSCs) (Issigonis et al, 2009). Strikingly, scRNA-seq data showed enrichment of metabolic pathways like oxidation–

---

Upper row: time-dependent size of individual ISC clones (different colors). Monoclonal conversion (red clone) takes place after about 16 and 8 wk in the crypt of the young and old mouse, respectively. Lower row: snapshots of the crypts at the time points indicated; SCs = stem cells, ECs = enterocytes, PCs = Paneth cells, GCs = goblet cells. **(F)** Box plots of the conversion times for strong (100%), medium (95%), and weak (90%) ISC–BM attachment strength in crypts of young mice (reference) and for weak adhesion (90%) in crypts of aged mice. N indicates the number of conversion events considered. The median conversion time of crypts from young mice (strong adhesion) and aged mice (weak adhesion) differs significantly (*** = $P < 0.001$). **(G)** Increasing fraction of monoclonal crypts over time for the same scenarios (clone size >3 cells). About 80% of the crypts of young mice with strongly adherent ISCs became monoclonal within 14 wk. Weak ISC–BM attachment in aged mice reduces this time to less than 8 wk.

reduction processes enriched in aged ISCs. Metabolic stress is another strong cofactor effecting crypt epithelial biomechanical properties such as adhesion (Smith et al, 2017). Our ISC aging gene expression signature identified by deep-learning methods showed a significant negative enrichment of the canonical ISC stem cell signature, which implies a negative correlation between aging and stemness in ISCs. Changes of competition dynamics have been investigated in mice (Snippert et al, 2014) and by applying single cell–based computational models (Thalheim et al, 2016). Although experiments allow precise manipulation, for example, of gene expression, manipulation of cell function is frequently impeded by regulatory redundancy. Here, single cell–based modeling supported an understanding of the link between ISC adhesion and tissue organization. The model results are in excellent agreement with our experimental findings and are very similar to changes in clonality obtained upon reducing Wnt secretion in cancer (Huels et al, 2018). In summary, the analysis of clonal dynamics in vivo supports an accelerated clonal drift upon aging, likely because of reduced adhesion of aged ISCs because of reduced canonical Wnt signaling.

# Materials and Methods

### Mice

C57BL/6 mice were bred in Tierforschungszentrum (TFZ), Ulm University. Tg(Vil-cre/ERT2)23Syr, *Gt(ROSA)26Sor^tm1(CAG-Brainbow2.1)Cle*/J, and B6.129P2-Lgr5^tm1(cre/ERT2)Cle/J were purchased from the Jackson Laboratory. Tg (Vil1-cre/ERT2) mice were crossed with Confetti; R26 to get vil1-cre; R26 Confetti mice and Lgr5EGFP-ires-creERT2 were crossed with Confetti; R26 to get Lgr5e-creERT2; R26 Confetti. All transgenic mice were backcrossed to C57BL/6 mice. To initiate the confetti fluorophores either in the whole intestinal epithelial cells or specifically in the Lgr5 cells, vil1-cre; R26 Confetti or Lgr5e-creERT2; R26 Confetti mice, respectively, were injected intraperitoneal with a single dose (5 mg) of tamoxifen (Sigma-Aldrich). All mice were housed in the animal barrier facility under pathogen-free conditions at the Ulm University. All mouse experiments were performed in compliance with the German Law for Welfare of Laboratory Animals and were approved by the Institutional Review Board of the Ulm University and by the Regierungspraesidium Tuebingen (state government of Baden-Württemberg), protocol number: 35/9185.81-3/1407. Housing conditions: a temperature range of 22°C ± 1°C, a relative humidity of 55% ± 10%, an air change rate of 15 times, and a light/dark change of 12/12 h. Nutrition: ssniff M-Z autoclavable complete for mice breeding (#V1124-3).

### Whole-mount tissue preparation for confocal imaging

The intestinal duodenum was harvested into ice-cold PBS, flushed with ice cold PBS, opened lengthwise, and fixed overnight in 4% paraformaldehyde (Sigma-Aldrich) at 4°C. Mucous and as much as possible of muscles lyres were removed bay a razor. The tissue was cut into 1-cm pieces and mounted on a slide with VECTASHIELD Antifade Mounting Medium (Vector Laboratories).

### Tracing and quantification of the crypt clones of vil1-cre; R26 confetti mice

Z-stacks images of whole-mount intestinal tissues were taken by Zeiss LSM 710 confocal microscope. For vil1-cre; R26 Confetti, crypt clones (each group of cells labeled with one color considered as one clone) were traced and analyzed from the bottom to the top for each individual crypt using Volocity image analysis software (v6.2; Perkin Elmer).

### Tracing and quantification of the crypt clones of lgr5-cre; R26 Confetti mice

Z-stack images were screened for individual clones labeled by YFP that did not overlap with other clones. First, the vertical extension of selected clones was analyzed. Clones that extended over the crypt orifice were rejected. Afterward, we calculated the area covered by YFP signal in all stacks using a Fiji-based (Schindelin et al, 2012) in-house plugin. The average area was calculated dividing the total area by the number of stacks over which the analyzed clone extends.

### Flow cytometry and cell sorting

To prepare single Lgr5 cells, crypts from Lgr5e-creERT2 mouse were isolated by incubating the opened lengthwise small-intestine pieces in the Gentle Cell Dissociation reagent (STEMCELL Technologies) on a rocking platform, at 20 rpm for 15 min at RT. Next the tissue pieces were resuspend in 10 ml ice-cold PBS containing 0.1% BSA and pipette up and down to strip off the crypts. The suspension was filtered through a 70-$\mu$m cell strainer. Filtered crypts were resuspended in 5 ml of TryPLE Express (Gibco) supplemented with Y27632 (10 $\mu$M) (Sigma-Aldrich). Crypts were transferred into a C-tube and dissociated by pre-programmed GentleMACS (Miltenyi Biotec). Cell suspension of the dissociated crypts were centrifuged and resuspended in 1 ml ice-cold DMEM/F-12 with HEPES (15 mM) (Gibco/Life Technologies) supplemented with N-acetylcysteine (0.5 mM) (Sigma-Aldrich), Y27632 (10 $\mu$M) and 1% (wt/vol) BSA. The cell number was counted by a cell counter to obtain a concentration of 1–5 × 10$^6$ cells/ml. Cells were stained with 7-AAD (BD Biosciences) and Annexin V (BioLegend). Lgr5 cells were sorted through a 100-$\mu$m nozzle using BD FACS Aria III (BD Bioscience) into DMEM/F-12 with HEPES (15 mM), supplemented with N-acetylcysteine (0.5 mM) and Y27632 (10 $\mu$M).

### Single-cell RNA-Seq analysis

Single, Lgr5$^{GFPhigh}$ cells (a total of 1,920 single Lgr5$^{GFPhigh}$ cells from 3 young and 4 old mice [young1 #9327 = 192 cells, young2 #9394 = 384 cells, young3 #9529 = 384 cells, old1 #7082 = 192 cells, old2 #7083 = 192 cells, old3 #7084 = 192 cells and old4 #7087 = 384 cells]) were sorted in 384-well low-bind plates (Eppendorf) containing 1.2 $\mu$l lysis buffer (0.1% Triton [Sigma-Aldrich], 1 U/$\mu$l RNase Inhibitor [Takara Bio], 2.5 $\mu$M polyT primer [Thermo Fisher Scientific], and 2.5 mM dNTPs [Thermo Fisher Scientific]). After sorting the plates, they were spun down and immediately frozen at −80°C. After thawing, the lysates were incubated for 3 min at 72°C and cooled on ice. A total of 2 $\mu$l RT

mix containing 20 U Maxima RT (Thermo Fisher Scientific), 1× RT buffer, RNAse Inhibitor (Takara), 11% PEG-8000, and 4.5 $\mu$M TSO (Eurogentec) was added. cDNA was generated for 90 min at 42°C. After inactivation at 70°C, 3 $\mu$l PCR mix was added containing 2× KAPA HS RM (Roche) and 0.2 $\mu$M IS PCR primer. cDNA was amplified for 21 cycles. cDNA was purified with AMPure beads using the Agilent Bravo platform with a bead to cNDA ratio of 0.8. QC was done from the selected wells, and cDNA diluted to 0.3 ng/$\mu$l. Libraries were prepared with the Illumina Nextera Kit using a customized downscaled version with the TTP labtech mosquito liquid handling system. Each position was individually indexed during the final PCR and pooled afterward. The pool was purified with AMPure beads (bead to library ratio was 0.9) followed by a QC with Qubit and TapeStation. Libraries were sequenced on a NextSeq550 with 384 cells per lane. Single-end reads were mapped to the mouse reference genome GRCm38 using HISAT2 (Pertea et al, 2016; Kim et al, 2019; Zhang et al, 2021). RNA-seq mapping QC was performed using Picard tools (https://broadinstitute.github.io/picard/). Downstream analyses including graph-based clustering, heatmaps, principal component analysis, and diffusion maps were conducted using R and Bioconductor packages (Gentleman et al, 2004; Huber et al, 2015). Deep-learning analysis was performed using the keras/TensorFlow (Abadi et al, 2016) (software available from tensorflow.org) implementation in R. GSEA was performed using the standalone Java implementation of the package (Subramanian et al, 2005). Pathway network analysis was performed within GSEA, and the plot was generated using Cytoscape (Shannon et al, 2003; Otasek et al, 2019).

## Crypt transduction with the barcode virus library and DNA harvest and amplification

Barcode library design and construction have been described elsewhere (Thielecke et al, 2017). A total of 200 fresh crypts from young (12–14 wk) and aged (more than 100 wk) mice were isolated as described above, mixed in suspension with the barcode lentivirus library with an MOI of 10, and incubated at a 37°C, 5% $CO_2$ for 12 min, with a gentle mixing in the middle time of incubation. The whole suspension was then mixed with 50 $\mu$l of Matrigel (Corning), plated in 24-well plate, and incubated in a 37°C, 5% $CO_2$ for 10–15 min until Matrigel polymerized. About 500 $\mu$l of complete IntestiClut TM medium (STEMCELL Technologies, supplemented with CHIR99021 [2.5 $\mu$M; Stemgent] and thiazovivin [2.5 $\mu$M; Stemgent]) was added and changed every 3 d. After 7 d, the whole organoids were harvested and dissociated with Gentle Cell Dissociation reagent on a rocking platform at 20 rpm at RT for 10 min. The first DNA patch was extracted from the dissociated organoids according to NucleoBond Xtra Maxi Plus kit instructions (Genetimes ExCell). The integrated barcode sequences were amplified from the genomic DNA, using indexed primers with Illumina adapters. Using Platinum SuperFi Green PCR Master Mix (Invitrogen), amplification was carried out for 40 cycles. PCR products were purified using Agencourt XP beads (Beckman Coulter). Purified PCR products were analyzed and quantified in Agilent Bioanalyzer. Then the PCR products were pooled and subjected to next-generation sequencing in Mi-Seq, for single-end 83-bps sequencing with addition of PhiX DNA.

## Data analysis barcode sequencing

After demultiplexing, the NGS results were analyzed using a published R-package (Thielecke et al, 2020). Single reads were quality checked (Phred-Score ≥ 30), barcode (BC) abundancies were calculated, and an error-correction was applied to cope with PCR- and NGS-introduced nucleotide exchanges. To use the benefits of the employed annotated barcode library (ABC library [Thielecke et al, 2017]), the standard error-correction method was adapted by focusing exclusively on the known barcode sequences (white list). After the extraction of all unique sequences and their abundances, the Hamming distances (HD) of all unknown sequences to all detected white-list BCs were calculated. Sequences within a HD of eight were considered a descended BC, and therefore, the number of reads was added to the read-count of the "original" white-list BC, and the sequence information was dismissed. Finally, only BCs with a read-count ≥100 were selected for further analyses.

## Determination of ISCs adhesion ability under shear stress

ISCs were sorted as described above; 0.5 or 0.2 × $10^5$ ISCs from young or old were suspended in 100 $\mu$l complete IntestiClut TM medium supplemented with CHIR99021 (2.5 $\mu$M) and thiazovivin (2.5 $\mu$M). Wnt3a was provided at 100 ng/ml. For the adhesion assay, ISCs were then cultured at 37°C, 5% $CO_2$ overnight in ibidi 0.4 Luer slides (ibidi) previously coated with coated with collagen type IV and subsequently treated by a thin layer of diluted Matrigel (1:125) in DMEM/F-12 with HEPES for 1 h at room temperature. Afterward, shear stress in the channel was increased by 1 dyne/$cm^2$ every minute from 1 to 33 dyne/$cm^2$ according to the manufactures guidelines via an ibidi pump system (ibidi) with a perfusion set (ibidi). Cells on the slide were observed on an inverted light microscope (Olympus IX83; Olympus), and adherent cells recorded in phase contrast every minute for up to 30 min using an Olympus LUCPlanFL N 20× and the Olympus microscope software. Analysis was performed using cellSens software (Olympus).

## Modeling of ISC competition within the crypt

The basic structure of our computational model of the small intestinal crypt of mice has been described in detail elsewhere (Buske et al, 2011; Thalheim et al, 2016). In short, the basic model comprises 3D representations of both the cells of the intestinal epithelium and the basal membrane (BM, see below). Cells are capable of forming contacts to neighbor cells and the BM. They can move, grow, and divide. Their fate is regulated by extrinsic activation of Wnt and Notch signaling. The activities of these pathways control cell proliferation and lineage specification. Stem cells (SCs) are characterized by active Wnt and Notch signaling (Wnt: high, Notch: high). The model considers specification of these cells into enterocytes ECs (Wnt: low, Notch: high) and two types of secretory cells: Paneth cells (PCs, Wnt: high, Notch: low) and goblet cells (GCs, Wnt: low, Notch: low). Cellular Wnt activity is assumed to depend on the cell's position P along the crypt–villus axis, decreasing with the distance from the bottom of the crypt. Thus, a SC that moves up the crypt axis decreases Wnt signaling and specifies into an EC at a threshold position P1. Cellular Notch activity is determined by the cell's neighbors. Secretory cells (PCs, GCs) activate Notch signaling

**Table 1. Parameter set of the reference crypt model.**

| Symbol | Value | Parameter | References |
|---|---|---|---|
| Parameter of the cell and cell–cell interaction model (set I) | | | |
| $V_0$ | $4/3\pi\,(R_0)^3$ | Minimal volume of an isolated cell | Estimated, $R_0$ = 5 $\mu m$ |
| E | 1 kPa | Young modulus | Galle et al (2005) |
| N | 1/3 | Poisson ratio | |
| $\varepsilon_c$ | 200 $\mu N/m$ | Cell–cell anchorage | |
| Parameter of the BM and cell–BM interaction model (set II) | | | |
| $z_0$ | 150 $\mu m$ | Length of the tissue segment | Buske et al (2011) |
| $r_0$ | 60 $\mu m$ | Upper radius of the tissue segment | |
| $\lambda_1$ | 0.25 | Shape parameter 1 | |
| $\lambda_2$ | 0.1 | Shape parameter 2 | |
| $\rho_0$ | $1.4/\mu m^2$ | Average knot density of the BM | Minimum to guaranty smooth cell motion |
| $\Omega$ | 0.95 R | Optimal cell–BM distance in cell radii R | Set |
| $\varepsilon_{rep}$ | 0.1 pNm | Cell–knot repulsion energy | Avoids BM penetration |
| $\varepsilon_{adh}^{SCPC}$ | 17.5 pNm | Cell adhesion energy of SCs and PCs | Enables stable adhesion without cell–BM linkage |
| $\varepsilon_{adh}^{ECGC}$ | 2.75 pNm | Cell adhesion energy of ECs and GCs | Fit: apoptosis rates <5% (Marshman et al, 2001) |
| Parameter of the cell growth and motion (set III) | | | |
| T | 14 h | Average cell growth time | Fit: effective cell cycle time ~24 h |
| $V_p$ | 0.88 $V_0$ | Threshold volume for contact inhibition | Set |
| $\eta_c$ | $5 \times 10^{10}$ $Ns/m^3$ | Friction constant for cell–cell friction | Galle et al (2005) |
| $\eta_{VO}$ | 400 Ns/m | Friction coefficient regarding volume changes | |
| $\eta_{BM}$ | 3.2 Ns/m | Friction coefficient for cell–BM friction | Fit: turnover |
| $\overrightarrow{F_A^{PC}}$ | –7.5 nN $\overrightarrow{e_z}$ | Active migration force of PCs | Fit: distribution of PCs |
| $\overrightarrow{F_A^{ECGC}}$ | 2.5 nN $\overrightarrow{e_z}$ | Active migration force of all other cells | Fit: Brdu data |
| Parameter of the cell fate model (set IV) | | | |
| $P_1$ | 21 $\mu m$ | Position of the Wnt threshold TW1 | Size of the niche |
| $P_2$ | 62.5 $\mu m$ | Position of the Wnt threshold TW2 | Fit: Brdu data |
| $N_1$ | 3 | Number of Notch ligand–expressing neighbors required for SC maintenance | Thalheim et al (2018) |
| $N_2$ | 1 | Number of Notch ligand–expressing neighbors required for EC maintenance | |
| $\tau^P$ | 8 wk | Average PC lifespan with contact to SCs | Thalheim et al (2016) |
| $\tau^A$ | 12 h | Max. PC lifespan without contact to SCs | |
| $\tau^{apop}$ | 0.36 min | Lifespan of a cell without contact to the BM | 1 simulation time step |
| Parameters of the cell–BM linkage in the SC niche (model extension, setV) | | | |
| $E^P$ | 0.2 mN/m | Elastic modulus of the polymer springs | Similar to cell–cell linkage of differentiated ECs in organoids (Thalheim et al [2018]) |
| $K^P$ | 10 fNm | Bending modulus of the polymers segments | |
| $l_{max}^P$ | 5.7 $\mu m$ | Cell–BM distance for cutting BM linkage | Adjusted to obtain the observed average monoclonal conversion time |
| $r_{adh}$ | 1/3 h | Update rate of the linkage | |
| $V_{max}^P$ | 1.85 $V_0$ | Threshold cell volume for cutting BM linkage | S-phase volume |

in neighboring cells via ligand–receptor interactions, although they themselves are not capable of activating the pathway. Thus, an SC that loses PC contacts specifies into a PC becaue of reduced Notch signaling. In a similar process, ECs that lose GC contacts specify into GCs. The model assumes thresholds (C1 and C2) for the number of PC and GC contacts required for a stable SC and EC fate, respectively. All fate decisions are assumed to be reversible as long as the cells are not terminally differentiated. This state is achieved if Wnt signaling falls below a second threshold value at position P2. As an exception, PCs become terminally differentiated at high Wnt

**Table 2. Parameter changes in the aged crypt model.**

| | | | |
|---|---|---|---|
| $z_0$ | 165 μm | Length of the tissue segment | Nalapareddy et al (2017), ensures constant number of intestinal stem cells |
| $\lambda_2$ | 0.3 | Shape parameter 2 | |
| $l^p_{max}$ | 0.95 $l^p_{max}$, 0.90 $l^p_{max}$ | Cell–BM distance for cutting BM linkage | Medium adhesion strength weak adhesion strength |

activity, after having finished a last cell cycle subsequent to specification. As long as specified cells are not terminally differentiated, they are considered as progenitors and thus are capable of proliferation. During proliferation, cells increase their volume with a defined growth rate until they reach twice their initial volume V0 and divide into two cells. Afterward, the daughter cells start increasing their volume again if they are not subject to contact inhibition, that is, are not compressed below a threshold volume Vp. Cell movement is determined by adhesion, deformation, and compression forces exerted by neighboring cells or the BM. In addition, ECs and GCs actively move up the crypt–villus axis and PCs down this axis. Cells that lose substrate contact undergo anoikis and are removed from the system. Cells that leave the crypt ($P > 30$) are removed as well. The reference parameter set applied in our simulations is given in Table 1.

In the standard model, cell–substrate adhesion, assumed to be proportional to the area of cell surface-BM contact, ensures that cells remain attached to the basal membrane, while being able to slight on it easily. Cell movement is controlled by cell–cell and cell–BM friction. In the extended model used in this manuscript, we represent the matrix by a dense polymer network and introduce an additional adhesion structure of niche cells that attaches these cells to individual matrix polymers (Table 2). Thus, cell movement in the niche requires a turnover of this adhesion structure; including rupture of bonds and formation of new bonds. The extended adhesion model thus incorporates an additional structure that links niche cells locally to the basal membrane. This extension is enabled replacing the continuous BM surface of the basic model by a dense polymer network as introduced for a model of intestinal organoids (Thalheim et al, 2018). The polymer network resamples the crypt shape, which is, in contrast to the organoid model, fixed. The cell center of niche cells (SCs and PCs) is linked by an elastic spring of intrinsic length $l_0$ to all knots of the network that are closer than $l_{max}$ to the cell center just after birth. This structure is updated during the cell cycle with rate $r_{adh}$. During this update, springs expanded above $l_{max}$ are removed, and new springs are built to all knots of the network that have become closer than $l_{max}$. The function of this structure is to dampen the cell movement in the niche on short time scales $t < 1/r_{adh}$.

### Statistical analysis

If not indicated otherwise, a *t* test or two-way ANOVA (Graph Pad Software) was used to assess the statistical significance of differences.

## Data Availability

Primary data will be provided upon request. Single-cell sequencing data on young and aged ISCs are deposited under the accession number GSE200536.

## Supplementary Information

## Acknowledgments

We would like to thank the TFZ of Ulm University and the Flow Sorting Core of Ulm University for their support. A Hageb was supported by a fellowship from the GRK Heist (DFG) and a short-term fellowship of the Professor Dieter Platt Foundation. J Galle/T Thalheim: the study was supported by the Bundesministerium für Bildung und Forschung (grant: INDRA, grant number: BMBF 031A312). The model computations were performed using a Bull/Atos cluster at the Center for Information Services and High Performance Computing (ZIH) at TU Dresden.

### Author Contributions

A Hageb: conceptualization, data curation, investigation, visualization, methodology, project administration, and writing—original draft, review, and editing.
T Thalheim: methodology, and writing—original draft, review, and editing.
KJ Nattamai: formal analysis and investigation.
B Mohrle: conceptualization and investigation.
M Sacma: formal analysis and investigation.
V Sakk: formal analysis and investigation.
L Thielecke: writing—original draft, review, and editing.
K Cornils: investigation.
C Grandy: formal analysis and investigation.
F Port: formal analysis and investigation.
K-E Gottschalk: conceptualization and investigation.
J-P Mallm: formal analysis and investigation.
I Glauche: formal analysis.
J Galle: methodology and writing—original draft, review, and editing.
MA Mulaw: formal analysis, investigation, and writing—original draft, review, and editing.
H Geiger: conceptualization, data curation, supervision, investigation, methodology, project administration, and writing—original draft, review, and editing.

### Conflict of Interest Statement

The authors declare that they have no conflict of interest.

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
