## [Reviewer comments · Life Science Alliance]

Life Science Alliance

Reduced adhesion of aged intestinal stem cells contributes to an accelerated clonal drift

Hartmut Geiger, Ali Hageb, Torsten Thalheim, Kalpana Nattamai, Bettina Möhrle, Mehmet Sacma, Vadim Sakk, Lars Thielecke, Kerstin Cornils, Carolin Grandy, Fabian Port, Kay-Eberhard Gottschalk, Jan-Philipp Mallm, Ingmar Glauche, Joerg Galle, and Medhanie Mulaw

DOI: <https://doi.org/10.26508/lsa.202201408>

Corresponding author(s): Hartmut Geiger, University of Ulm and Medhanie Mulaw, Institute of Experimental Cancer Research

Review Timeline:

Submission Date:	2022-02-14
Editorial Decision:	2022-02-14
Revision Received:	2022-04-06
Editorial Decision:	2022-04-06
Revision Received:	2022-04-11
Accepted:	2022-04-12

Transaction Report:

Please note that the manuscript was previously reviewed at another journal and the reports were taken into account in the decision-making process at Life Science Alliance.

Referee #1 Review

Report for Author:

The manuscript by Hageb and colleagues documents accelerated clone dynamics in aged murine intestinal stem cells that motivates a number of detailed analyses aiming to explain how this is mediated. The topic is clearly relevant to age related intestinal dysfunction and to age related disease, notably cancer and potentially of interest to the readership of this journal.

The authors start by showing accelerated intracrypt clone dynamics in aged (80-85 weeks) compared to young (12-16 weeks) mice following a single treatment with tamoxifen to activate a confetti reporter cassette. One immediate and important issue is the identity of the Cre driver line. That stated, Tg(Vil1-cre)^{1000Gum/J}, does not require induction obtain expression of active Cre recombinase. The initial results do seem to show that there are more monoclonal and fewer bi/multiclonal crypts in older mice compared to young mice across the time course. However, the authors make no attempt to compare their time course for young mice to that described in the literature. They show only around 40% of crypts to be monoclonal at 8 weeks post induction whereas published literature would suggest that the frequency should be around 75%- 85% (of the different publications describing the drift phenomenon Snippert et al, 2010 is probably the most relevant as it used the Confetti allele employed here. Relatedly the statement in Introduction that the time to monoclonality is 30 weeks is incorrect and not supported by the references given (Griffiths et al. 1988; Winton and Ponder 1990; Li et al. 1994; Snippert et al. 2010; Lopez-Garcia et al. 2010), noting that Li et al is not included in references).

Given robustness across the different studies this is perplexing. Conceivably baseline clone dynamics could be influenced by environment and diet. The authors should specify these in Methods.

Attempting to build on the young vs old difference in intracrypt clone dynamics the authors investigate if clonal expansions to form multicrypt patches are also altered in old mice by looking at non-randomness in the distribution of adjacent crypts of the same confetti color. Here the use of a single metric - Median absolute deviation - to define non-random behaviour in older mice seems flimsy. What arises from different starting conditions, and what develops due to crypt fission subsequently? Some testing of predictions is necessary. The non-randomness is interpreted as the expansion of a few dominant clones in aged mice. The prediction then is that patch size distributions should show a shift with more monoclonal crypts and a small number of larger clones in old mice compared to young mice. Yet the latter analysis has not been done. Another paper by Snippert et al (2014)

uses confetti to infer a homeostatic fission rate in (presumably young mice) of around 5% per month. This would have been a useful benchmark to determine if young mice in this study show a similar fission rate and if that of older mice is skewed with respect to outlier patch sizes. The authors interpretation "our data demonstrate that upon aging crypts still show a clonal drift and clonal succession, but with a very much accelerated turnover of the dominant clone compared to young crypts, while the clonality is also more likely to spread among aged crypts." is not a sufficient explanation of what might be happening. Presumably most old crypts/stem cells are showing the accelerated dynamics, but only a few crypts/stem cells are dominant in mediating larger expansions. What is the unifying hypothesis here?

Minor point

Results para2: We therefore determined whether neighboring crypts with the same color were stochastically distributed or in distributed as clusters, which would indicate that drift might affect multiple crypts (SFigure 1c).

Clumsy sentence, drift affects all crypts, what are the authors trying to say?

Crypt fission and fusion is common in ISC biology (Bruens et al, 2017)

Common as a description of these phenomena is not helpful in terms of resolving if these processes are relevant to the author's observations. Indeed, if fusion is common in young mice and less common in old mice that might explain the differences in monoclonal and multi/biclonal crypts shown in Figure 1.

Fig1 legend: $*=p<0.05$, two-way ANOVA.

This description doesn't seem to capture the comparisons made in the figures where time points are compared with more than one asterix. And how are the error bars derived?

The next section identifies that clones are the same size, occupy the same vertical depth and cover the same area of crypt circumference in both young and old mice. This section initially seems straightforward and the author's interpretation is a relatively negative one: "The findings excludes heterogeneity in proliferation an expansion upon aging as a central mechanism for the accelerated drift upon aging"

However, this has to be reassessed in light of the later comment (SFig5 and related text) that aged crypts are larger and that the author's have described previously. It is unclear whether similarities in stem cell representation and clone behaviours are relative or absolute. The authors should make some effort to guide the reader here. For example, if crypts occupy a greater area then are the area measurements in Fig2c absolute area measurements. Fig2c is in any case hard to interpret, it shows data but no summary metrics are extracted to formally demonstrate there really is 'no difference'.

Minor points

Fig2f; The authors point to similarity in this figure showing clone representation but the 5 day comparison of frequency of confetti labelling in young vs old crypts indicates a reduction in the former; so young mice are more likely to lose clones- this doesn't seem to fit with decreased adhesive properties in old stem cells as proposed later.

Organoid barcoding experiments showing similar clonal complexities in young and old stem cells seem well performed.

The single cell sequencing approach certainly seems a reasonable way forward to assess heterogeneity in abundance of transcripts between stem cells from young and old mice. This is done by clustering cells from individual animals and identifying an average of 4 such clusters over 7 animals. The assessment of heterogeneity within stem cell clusters ("that aged ISCs formed more compact and distinct clusters comprising cells with high transcriptional similarity within a cluster but more distinct from cells in other clusters") is probably correct, but it seems a particular manifestation of heterogeneity the functional significance of which is unclear. Moreover, the stated objective was to find an aging related gene expression signature (page 7 penultimate line) yet although found and refined there is no actual signature presented. How many genes are in the 5% of the gene signature that give the highest 'likelihood predictors' and what are they? Also there is no attempt to relate the clusters to the biology of the tissue leaving many open questions that create uncertainty about what they represent. Projecting all the cells onto a common UMAP and being able to see how the clusters relate to stem cells occupying naïve, primed or committed states would have reassured. How many unique clusters would come from analysis of the pooled population? Do common colors assigned to clusters from different mice indicate similarity in clusters called between mice? Are 1920 cells sufficient to do this analysis as on average each cluster from each mouse is supported by only around 68 cells?

Guided by gene set enrichment analysis that identified cell adhesion as an altered GO term motivated experiments looking at adhesion of young and old stem cells under conditions of increasing flow stress. Appropriately older stem cells seem less adhesive as predicted. This was related to reduced Wnt signaling in older stem cells and the adhesion of older stem cells partly restored by addition of exogenous wnt3a. The experiments seem fine. And their computational imputation of the effects of reduced adhesion shows that this might explain the clonal behaviours described earlier. The uncertainties come in the absence of any exploration of what is happening in vivo in terms of available substrates and mediators of epithelial stem cell adhesion being altered with age. Even accepting the generic interpretation, Wnt signalling is implicated in commitment and maturation processes. Returning to the identity of the different clusters from the single cell sequencing, is there an increased proportion of primed/committed cells expressing Lgr5 (the marker defining stemness here). Also see above comment relating to Fig2f.

If stem adhesion is decreased then one might imagine that the rate of stem cell loss is increased but that this is compensated for by increased replacement rates of surviving stem cells. In which case are adhesion effects so easily divorced from expansion?

Overall, the manuscript presents observations that are individually of interest but that do not link in a persuasive way to build a convincing story. Their initial hypothesis and final interpretation are not precisely defined. The clonal behaviours in young and old mice are not comprehensively explored. The descriptions of stem cell changes with age are conveyed qualitatively making their magnitude and their biological significance difficult to appreciate. The authors need to focus on understanding the clonal behaviours in young and old mice more fully in order to convince that adhesion effects can solely explain age related differences.

Referee #2 Review

Report for Author:

The manuscript entitled 'Reduced adhesion of aged intestinal stem cells contributes to an accelerated clonal drift' by Hageb et al. suggests a role for the strength of ISC adhesion to the niche in determining the speed of crypt fixation. The authors claim that aging results in reduced adhesion, thereby increasing the speed crypt fixation. Although this idea is highly interesting, the authors do not fully provide the experimental evidence to support their claims. In particular, the manuscript mostly confirms existing work on ISCs and aging using the well-studied Confetti mouse model, and the novel findings on cellular adhesion are very limited. This leads to the following major concerns:

1. This study demonstrates how aging increases the rate of crypt fixation in young versus old mice using the Confetti mouse model. This finding is not novel since several studies have revealed how crypt fixation is increased when the number of functional ISCs per crypt decreases, e.g. by decreasing Wnt signaling. Moreover, various studies have demonstrated that aging results in reduced ISC fitness by a decrease in Wnt signaling, which is caused by both cell intrinsic signaling and by altered secretion of niche factors, thereby generally outlining the mechanism of increased crypt fixation in aged animals. Moreover, it has been shown that addition of Wnt3a or inhibition of Wnt reducing factors rescues the effect of decreased ISC fitness, and it should therefore come as no surprise that in the current study addition of Wnt3a rescues the loss of attached cells in vitro.
2. The authors claim it is not necessarily decreased number of ISCs that causes increased crypt fixation, but a decrease in cellular adherence of ISCs. They base their conclusion on results obtained using an Lgr5-GFP mouse model, and demonstrate that the number of Lgr5-GFP cells stays equal in young versus old mice. However, as the authors pointed out in the introductory section, the number of Lgr5+ cells does not reflect the number of functional ISCs, while the number of functional stem cells dictate the rate of crypt fixation. In addition, it remains unclear why the authors use the Lgr5 mouse model in figure 2 with the sole purpose of quantifying vertical extension, whilst this does not necessarily say something about (horizontal) ISC competition within the crypt bottom.
3. The authors use scRNA-sequencing of Lgr5-GFP^{high} cells of young and old mice to observe whether there are any differences between young and old ISCs. They identify 4 distinct clusters within the Lgr5-GFP^{high} population in both young and old cells, and claim that there is increased heterogeneity within the old ISC population. However, given the fact that the same 4 clusters are identified in the young and old sample, and the complete lack of characterization of these clusters, the authors cannot make such solid claims. Moreover, it is unclear if the reduced biological adhesion as shown in figure 4d accounts for all 4 clusters in the 'old' sample, or whether a single cluster shows a lack of adhesion.
4. Although the decrease in cellular adhesion as found using scRNA-seq is the most novel and interesting part of the study, the authors have not provided any data to functionally validate this finding to provide any mechanistic insight into this phenomenon. More specifically, it is unclear which adhesion molecules are reduced, whether they are reduced in some old ISCs or all ISCs, and the presence of such adhesion molecules has not been visualized either in intestinal tissue sections or in vitro organoid cultures.
5. Furthermore, the authors demonstrate that old intestinal cells have reduced adhesive strength using an in vitro stress model, and that this reduction can be rescued by addition of Wnt3a. Based on these findings the authors claim that reduced adhesion is the result of decreased Wnt signaling. However, due to a lack of mechanistic insight, again the authors should not make such firm claims, as they have only demonstrated a sole correlation between Wnt signaling and adhesive strength. In addition, since their stress model does not reflect the architecture of the normal intestinal crypt (and probably also not the forces that are imposed on the crypt), the authors should validate their findings in an in vivo model. Moreover, to prove that cellular adhesion is a critical influencer of crypt fixation, the authors should modulate these adhesive strengths and demonstrate how this alters fixation rates.

Minor concerns:

- figure 2a is not mentioned in the text
- figure 3 b, c, and d, are unclear in terms of figure quality and interpretation

- the GSEA plots throughout the manuscript are hard to interpret because labels of the sample are missing on the X-axis.
- Figure 4h is impossible to interpret

Referee #3 Review

Report for Author:

The manuscript by Hageb and colleagues describes clonal dynamics of the intestinal stem cells in old and young mice. The authors investigate whether ISC's properties change with age using lineage tracing analyses. They find that certain stem cells are more dominant than the others with age. Based on single-cell RNA-sequencing data, the authors conclude that cell adhesion might be important for defining the properties of aged ISCs.

Major concerns:

The 3rd sentence of the Results part: the authors use B6.Cg-Tg(Vil1-cre)1000Gum/J mice for the lineage tracing analysis either in young or old mice. The same strain is described in the methods part. They use tamoxifen to induce the expression of fluorescent reporters. However, the indicated mouse strain is not Cre-ERT but only Cre. Therefore, all conclusions in Figure 1 about the difference in clonal dynamics between the young and old mice is not relevant.

Figure 2: how do the authors explain rather dim RFP fluorescence in this figure compared to very bright RFP in Figure 1b?

Figure 3: The sentence describing Figure 3b "Organoids formed from old crypts showed small difference in the frequency of samples in which a small number of barcodes dominated the relative abundance in old organoids" is incomprehensible. The same is true for the conclusion sentence: "This data is consistent with no or only a very minor advantage of individual aged ISCs in conferring a dominance upon ex vivo competition that is distinct from young ISCs, even when competing against ISCs from different crypts".

How could ISCs from different crypts compete ex vivo? What is this conclusion based on?

I do not see the data showing competition properties for the young ISCs.

The whole barcoding strategy/ outcome is not well described, and the graphs are not clear. Furthermore, ex vivo aged organoids do not have any competition advantage, which is in contradiction to the statement in the Summary.

Figure 4: The authors state that GSEA analysis showed a negative correlation with ISC signature published by Munoz et al.,. I assume if they sorted and sequences stem cells the correlation must be positive.

Figure 4d and 4e do not carry much meaning.

The gene expression differences between young and aged mice could be due to the difference in the regional identity, the anterior versus posterior, of the LGR5-EGFP ISCs. scRNA-seq UMAP or t-sne plots displaying the expression of the regional markers, such as Fabp1, Fabp2, Fabp5, Gata4, Gata6, Hmgb2, Olfm4 would be indicative that the ISCs are coming from the same regions. Furthermore, the expression of the certain markers for adherence junctions displayed as UMAP plots would be more convincing than GSEA plots. For example, GO-adherence junction- the enrichment score is equally high for either red (young?) and blue (old?). So, are they equal? And what are the genes included inside?

Figure 4F: What is Young_9529 and Old_7087? What are the 4 clusters? Why they are separated in clusters?

The conclusion of Figure 4 should be strengthened by showing more data, markers. Are those few old cells in black ISCs or +4/TA cells?

Figure S4F: Perhaps it is a PDF conversion, but I see that Tert and Tnfrsf19 are higher expressed in the old cells, yet the mean is lower in those cells. Could, please, the authors check for that? The same is for smoc2 and axin2, although less pronounced.

Figure 5a: Many genes assigned as adhesion are not encoding for adhesion molecules, such as Rgmb (L-glutamine:2-deoxy-scyllo-inosose aminotransferase), Bcl6 (transcription factor), Apc (cytoplasmic), C1qntf1 (I Beta-1,6-N-Acetylglucosaminyltransferase), etc.

If aged ISCs express lower levels of integrins (or other adhesion molecules) compared to the young ISCs, it should be shown using antibody stainings on tissue sections.

It is not clear whether the organoid assays for shearing stress were performed with CHIR or not. If with, when adding Wnt-3a (undescribed concentrations) would have no difference for the assay. As the amount of CHIR added is high enough to inhibit GSK3b to a very minimum.

The message about the involvement of adhesion in the clonal drift of ISCs is important, therefore, in vivo evidence, including either loss-of-function mouse models or small molecule inhibitors are desirable to confirm the computational predictions.

In summary, if the adhesion is important then the dominant aged ISCs should express the adhesion molecules at higher levels compared to the excluded aged ISCs. That is not shown by the authors. There could be a difference between the young and the old, but those ISCs do not meet each other in vivo.

February 14, 2022

Re: Life Science Alliance manuscript #LSA-2022-01408-T

Hartmut Geiger
University of Ulm
Department of Dermatology and Allergic Diseases, University of Ulm
James Franck Ring 11 C
Neues Forschungsgebäude, R.3010
Ulm 89081
Germany

Dear Dr. Geiger,

Thank you for submitting your manuscript entitled "Reduced adhesion of aged intestinal stem cells contributes to an accelerated clonal drift" to Life Science Alliance. We invite you to submit a revised manuscript addressing the Reviewer comments.

Thank you for this interesting contribution to Life Science Alliance. We are looking forward to receiving your revised manuscript.

Sincerely,

B. MANUSCRIPT ORGANIZATION AND FORMATTING:

Point-to-point response to the comments of the reviewers:**Referee #1:**

The manuscript by Hageb and colleagues documents accelerated clone dynamics in aged murine intestinal stem cells that motivates a number of detailed analyses aiming to explain how this is mediated. The topic is clearly relevant to age related intestinal dysfunction and to age related disease, notably cancer and potentially of interest to the readership of this journal.

Response: We really appreciate the very positive statement on the relevance and novelty of our study.

The authors start by showing accelerated intracrypt clone dynamics in aged (80-85 weeks) compared to young (12-16 weeks) mice following a single treatment with tamoxifen to activate a confetti reporter cassette. One immediate and important issue is the identity of the Cre driver line. That stated, Tg(Vil1-cre)1000Gum/J, does not require induction obtain expression of active Cre recombinase.

Response: We used the inducible mouse model for Vil-Cre induction, Tg(Vil-cre/ERT2)23Syr for all of the analyses in this study. We indeed listed a wrong strain in the manuscript text and also in the first section of the methods section on mice, and we need to really apologize for that mishap. The correct line is now listed in the text and in methods (lines 115 and 373).

The initial results do seem to show that there are more monoclonal and fewer bi/multiclonal crypts in older mice compared to young mice across the time course. However, the authors make no attempt to compare their time course for young mice to that described in the literature. They show only around 40% of crypts to be monoclonal at 8 weeks post induction whereas published literature would suggest that the frequency should be around 75%- 85% (of the different publications describing the drift phenomenon Snippert et al, 2010 is probably the most relevant as it used the Confetti allele employed here. Relatedly the statement in Introduction that the time to monoclonality is 30 weeks is incorrect and not supported by the references given (Griffiths et al. 1988; Winton and Ponder 1990; Li et al. 1994; Snippert et al. 2010; Lopez-Garcia et al. 2010), noting that Li et al is not included in references). Given robustness across the different studies this is perplexing. Conceivably baseline clone dynamics could be influence by environment and diet. The authors should specify these in Methods.

Response: The manuscript now includes, in the Discussion section, a more in-depth comparison of our data to data on monoclonal conversion to the literature (lines 318-24 and we agree with the reviewer that our summary of the current data was not fully adequate in terms of listing in more detail also the heterogeneity of the literature.

Our data on conversion in young animals for example is very consistent with data published by Lopez-Garcia et al. 2010 or with (Huels *et al*, 2018). Similar to us, Lopez-Garcia used whole mount tissue analyses and reported ~50% monoclonal crypts on week 8 of tracing, which is close to the ~46% seen in our experiments. Huels et al for example report the fraction of fixed (aka monoclonal) clones to be 50% at 7 weeks. Snippert et al used semi-thick sections, which might contribute to the difference in percentages listed by Snippert et al.. Within whole mounts, it is technically easier to detect even a single cell with another color, and which will then result in a scoring of bi-clonal in our experiments. Another explanation could be, as the reviewer already mentioned, environmental factors (microbiome, immune system, even diet), that might further influence dynamics. We discuss this also in lines 319-324.

Lines 319-324: "Our data on the dynamics of the drift in young animals is similar to the dynamics described by Lopez-Garcia et al (Lopez-Garcia *et al*, 2010) which also report close to 50% of all crypts being monoclonal 8 weeks after induction, while in our experiment

monoclonality is somewhat delayed when compared to data reported by Snippert et al. (Snippert *et al*, 2010), which could be due to differences in tissue analysis (whole-mount staining vs. thin-sections), or due to differences in environmental factors.”

All our animals (young and aged) were housed under identical conditions. We also added the additional information on housing and food to material and methods:
Lines 380-386: “All mice were housed in the animal barrier facility under pathogen free conditions at the Ulm University. All mouse experiments were performed in compliance with the German Law for Welfare of Laboratory Animals and were approved by the Institutional Review Board of the Ulm University as well as by the Regierungspraesidium Tuebingen (state government of Baden-Württemberg), protocol number: 35/9185.81-3 / 1407. Housing conditions: a temperature range of 22 +/- 1 °C, a relative humidity of 55 +/- 10%, an air change rate of 15 times and a light/dark change of 12/12 hrs. Nutrition: ssniff M-Z autoclavable complete for mice-breeding (# V1124-3)”.

We also modified sections in the Introduction part, lines: In young mice, reports support that in 7-8 weeks between 50% and up to 75-80% of all crypts turn monoclonal, while it might take up to 30 weeks to turn all crypts monoclonal (Li *et al*, 1994; Lopez-Garcia *et al.*, 2010; Snippert *et al.*, 2010; Winton & Ponder, 1990). During the next couple of weeks, one novel, again almost equipotent neutral ISC subclone within the currently monoclonal crypt, will replace the current dominant ISC clone to turn the crypt again monoclonal.

Attempting to build on the young vs old difference in intracrypt clone dynamics the authors investigate if clonal expansions to form multicrypt patches are also altered in old mice by looking at non-randomness in the distribution of adjacent crypts of the same confetti color. Here the use of a single metric - Median absolute deviation - to define non-random behaviour in older mice seems flimsy. What arises from different starting conditions, and what develops due to crypt fission subsequently? Some testing of predictions is necessary. The non-randomness is interpreted as the expansion of a few dominant clones in aged mice. The prediction then is that patch size distributions should show a shift with more monoclonal crypts and a small number of larger clones in old mice compared to young mice. Yet the latter analysis has not been done. Another paper by Snippert et al (2014) uses confetti to infer a homeostatic fission rate in (presumably young mice) of around 5% per month. This would have been a useful benchmark to determine if young mice in this study show a similar fission rate and if that of older mice is skewed with respect to outlier patch sizes.

Response: We provide indeed only one parameter to identify whether there is non-random distribution of crypts. The determination of the median absolute deviation though is widely accepted and a valid tool to score non-random behavior, which also includes predictions of a random distribution in the column “random” in SFigure 1d. The reviewer is correct that our data does not allow, as done by Snippert et al., to determine the fission rate, as we lack time dependent data in this analysis as well as patch size. The purpose of our analyses was not to quantify in detail fission or fusion, but rather the identification of data that might translate into such a novel hypothesis, that fission and fusion might be affected by aging. Whether the increase deviation of the old crypts from the young crypts is indeed due to changes in fission and fusion will need to be further investigated. Similarly, whether or not it is linked to the change in the crypt drift. Both, among other, are valid hypothesis.

The authors interpretation “our data demonstrate that upon aging crypts still show a clonal drift and clonal succession, but with a very much accelerated turnover of the dominant clone compared to young crypts, while the clonality is also more likely to spread among aged crypts.” is not a sufficient explanation of what might be happening. Presumably most old crypts/stem cells are showing the accelerated dynamics, but only a few crypts/stem cells are dominant in mediating larger expansions. What is the unifying hypothesis here?

Response: We share the view of the reviewer that our interpretation of the data is one valid explanation of our primary data. We also share the view that it might not be the only interpretation. We now separated the statement on the turnover of individual crypts from the spreading, as indeed, they might be two very distinct processes, and there might not be a unifying theory for both of them. Additional investigations will be necessary to determine this. This reads now the following in the manuscript (lines 145-149):

“In aggregation, our data show that upon aging crypts still show a clonal drift and clonal succession, but with a very much accelerated turnover of the dominant clone compared to the turnover in young crypts. In addition, clonality is also more likely to spread among aged crypts compared to young crypts. Whether both of these findings are mechanistically linked will need to be further investigated”.

Minor point

Results para2: We therefore determined whether neighboring crypts with the same color were stochastically distributed or in distributed as clusters, which would indicate that drift might affect multiple crypts (SFigure 1c).

Clumsy sentence, drift affects all crypts, what are the authors trying to say?

Response: We agree with the reviewer that similar to the question above, it remains unclear whether changes in clone dynamics and the changes that affect multiple crypts and are likely linked to fission/fusion are indeed linked or not. We thus simplified this sentence (lines 137-139): “ We also determined whether crypts with the same color were stochastically distributed or distributed as clusters, which would indicate that drift might affect multiple crypts”.

Crypt fission and fusion is common in ISC biology (Bruens et al, 2017)

Common as a description of these phenomena is not helpful in terms of resolving if these processes are relevant to the author's observations. Indeed, if fusion is common in young mice and less common in old mice that might explain the differences in monoclonal and multi/biclonal crypts shown in Figure 1.

Response: We agree that this sentence is misplaced in the logical flow, and thus added a modified version two sentences down to the manuscript. We also omitted the word common. “This implies that upon aging, there might be an accelerated spreading of dominant clones among neighboring crypts, which could be linked to the accelerated clonal drift but also to enhanced crypt fission and fusion upon aging (Bruens *et al*, 2017).” (lines 142-145)

This is also an interesting hypothesis put forward by the reviewer, which we will plan to test in future experiments (differences in fusion upon aging). As we see though larger patches monoclonality with the same color in aged mice, it is more likely that fusion is influence in a positive fashion.

Fig1 legend: *=p<0.05, two-way ANOVA.

This description doesn't seem to capture the comparisons made in the figures where time points are compared with more than one asterix. And how are the error bars derived?

Response: We now list in the figure legend the p-values that go along with multiple stars (**= p<0.01, ***= p<0.001, **** = p<0.0001). The error bars are defined, as further emphasized in the figure legend, as SEMs per data point. This section of the figure legends now reads the following (lines 695-703):

“Analysis have been performed on confocal z-stacks (each stack has 15-22 layers with 3.6 μm distance for each) covering the whole length of all the crypts analyzed. N = 3-5 mice (young or old for each time point analyzed). 3-5 stacks per animal and time point and on

average of ~ 80 crypts for each stack were analyzed. The data from all z-stacks from one animal were averaged to obtain a single value on clonality per mouse per timepoint. In total, 3437 (1890 young and 1547 old), 2627 (1188 young and 1439 old), 1644 (716 young and 928 old) and 1956 (780 young and 1176 old) crypts were analyzed for the 5d, 4 week, 8week and 20 week timepoint. Shown are means with SEM, **= p<0.01, ***= p<0.001, **** = p<0.0001, two-way ANOVA.”

The next section identifies that clones are the same size, occupy the same vertical depth and cover the same area of crypt circumference in both young and old mice. This section initially seems straightforward and the author's interpretation is a relatively negative one: "The findings excludes heterogeneity in proliferation an expansion upon aging as a central mechanism for the accelerated drift upon aging" However, this has to be reassessed in light of the later comment (SFig5 and related text) that aged crypts are larger and that the author's have described previously. It is unclear whether similarities in stem cell representation and clone behaviours are relative or absolute. The authors should make some effort to guide the reader here. For example, if crypts occupy a greater area then are the area measurements in Fig2c absolute area measurements. Fig2c is in any case hard to interpret, it shows data but no summary metrics are extracted to formally demonstrate there really is 'no difference'.

Response: We would like to first emphasize that data in Figure 2b,c is solely focused on individual ISCs in a crypt, and simply an analysis of the existing images (one ISCs and its offspring per crypt). In Figure 5S, we indeed analyze whole crypts, and the model determines competition among ISCs within a crypt. The reviewer is correct that whenever our data in Figure 2 is listed as μm or μm^2 , these are absolute measurements.

Having said that, the results of Figure 2b,c and S5a,b have been consistent. Figure 2b,c show properties of the ISC clones of young and old mice. Figure 2b shows the distribution of the vertical extension of the clones in μm , and Figure 2c the average area of the clones in μm^2 . Together these results characterize the absolute size of the clones.

Figure S5a provides information about the size of the crypts assumed in our simulations. So it is simulated data. In agreement with former publications, crypts of old mice are assumed to be larger than those of young mice for the simulation approach. Figure S5b shows simulation results for the vertical size distribution of ISC clones considering these differences. They nicely fit with the results provided in Figure 2b, except of some large clones found in the simulations that are not recorded in vivo because they extend into the region of the crypt-villus junction and cannot be imaged with enough confidence anymore in our sections.

We agree, that it may be difficult to interpret Figure 2c and that other type of presentation of the data might be better for the reader. In the original figure, we showed the properties of all individual clones analyzed. We now provide instead box plots for the clones of defined vertical extension (see below, novel Figure 2c), comparing clones of old and young mice. Also in this type of presentation (of the same primary data), there is not significant difference in clone size among young and aged ISCs.

Novel legend to novel Figure 2c (now lines 710-713): Average area covered by the YFP signal of clones in young and aged crypts 5 days post tam induction. Clones within defined ranges of vertical extension have been clustered (2 young and 2 old mice, between 60 and 90 clones per mouse). Few clones with an extension above 55µm were omitted.

Minor points

Fig2f; The authors point to similarity in this figure showing clone representation but the 5 day comparison of frequency of confetti labelling in young vs old crypts indicates a reduction in the former; so young mice are more likely to lose clones- this doesn't seem to fit with decreased adhesive properties in old stem cells as proposed later.

Response: We agree with the reviewer that this data (SFig2f) can be interpreted in different ways, and that it remains a scientifically challenging topic. The data in figure SFig2f only shows that the frequency of labeled crypts in young animals is, after 5 days, slightly reduced in young compared to old crypts. One explanation might be indeed that ISC's are quicker lost in young compared to old crypts, and one explanation for that could be in theory reduced adhesion of young ISC's. Our data though clearly shows reduced adhesion of aged, not young ISC's. That implies that likely other mechanisms are at play here, and there are indeed a good number of other explanations possible that are equally valid. We are simply currently not able, based on our data, to mechanistically understand the nature of this reduced frequency, and additional experiments will be required to determine the underlying mechanism. We are also not able to predict at the moment in as much a then identified mechanism will indeed be linked to the accelerated drift and the reduced adhesion.

Organoid barcoding experiments showing similar clonal complexities in young and old stem cells seem well performed.

Response: Thank you for this positive statement on the barcode experiments.

The single cell sequencing approach certainly seems a reasonable way forward to assess heterogeneity in abundance of transcripts between stem cells from young and old mice. This is done by clustering cells from individual animals and identifying an average of 4 such clusters over 7 animals. The assessment of heterogeneity within stem cell clusters ("that aged ISCs formed more compact and distinct clusters comprising cells with high transcriptional similarity within a cluster but more distinct from cells in other clusters") is probably correct, but it seems a particular manifestation of heterogeneity the functional significance of which is unclear. Moreover, the stated objective was to find an aging related gene expression signature (page 7 penultimate line) yet although found and refined there is no actual signature presented. How many genes are in the 5% of the gene signature that give the highest 'likelihood predictors' and what are they?

Response: We include now a supplementary table showing the list of the top 5% genes with the highest correlation coefficient to the prediction model (line 227). The PCA that we presented is based on this list. The list contains 278 genes including *Ly6e*, *Galnt12*, and *Tenm4* as top genes with positive correlation to aging (higher aging likelihood). On the other hand, genes like *Bcl6*, *Cpm*, and *Rps4l* were positively correlated with the young ISC phenotype (lower aging likelihood).

Also there is no attempt to relate the clusters to the biology of the tissue leaving many open questions that create uncertainty about what they represent.

Response: The sole goal and purpose of these single-cell analyses for this manuscript was to determine the transcriptional heterogeneity among ISCs upon aging in a completely unbiased fashion. The focus was not to analyze in general the ISC aging signature and likely deviations from that (naïve, primed, or committed). This would have needed to be performed along information of "pre-informed" signatures.

Projecting all the cells onto a common UMAP and being able to see how the clusters relate to stem cells occupying naïve, primed or committed states would have reassured. How many unique clusters would come from analysis of the pooled population? Returning to the identity of the different clusters from the single cell sequencing, is there an increased proportion of primed/committed cells expressing *Lgr5* (the marker defining stemness here). Also see above comment relating to Fig2f.

Response: The clustering has been performed independently of the aging or any other signature, and was only afterwards correlated to the aging signature (Figure 4h), which indeed identified that the aging signature is to a large extent determining the clustering in each individual mouse (Figure 4i, pseudo-time analysis). The focus was not to analyze in general the ISC aging signature and likely deviations from that (naïve, primed, or committed). This would have needed to be performed along information of "pre-informed" signatures. We plan to further use the pooled single cell information in a subsequent manuscript to present more in-depth information on the other aspects not related this manuscript on the single cell RNA sequencing data.

Additionally, once we identified a highly significant aging predictor genes (hence aging signature genes) using all cells across all individuals, we assessed, on the level of individual mice, the following points:

1. Do we see differences between young and aged mice in number of gene expression based clusters?

2. Are the clusters more/less defined in young vs. aged mice?
3. Do we see differences between clusters in a given mouse with respect to the aging signature that we identified?

Briefly, what we noted was there is no significant difference in number of clusters but the aged mice clustered were more defined and showed significant difference in the inter-cluster distance as compared to young mice. Additionally, diffusion map analyses revealed that there is no difference in the young mice clusters with respect to the aging signature while clusters in aged mice showed directionality and ordered difference based on the aging signature genes.

Do common colors assigned to clusters from different mice indicate similarity in clusters called between mice?

Response: Common clusters assigned to different mice do not indicate similarity among these clusters across animals. Each cluster remains unique and linked to the individual animal.

Are 1920 cells sufficient to do this analysis as on average each cluster from each mouse is supported by only around 68 cells?

Response: Our single cell data is based on SmartSeq2 technology, not on a 10x protocol. We obtain a much higher sequencing depth per cell with the SmartSeq2 technology compared to the 10x approach. We performed very thorough statistical analyses on the outcome of our data and the statement that upon aging, the difference among the clusters is increased. We thus can conclude that the number of cells was sufficient for the cluster analysis, while we agree with the reviewer that more cells will allow for an even more detailed determination of the structure of clusters.

Guided by gene set enrichment analysis that identified cell adhesion as an altered GO term motivated experiments looking at adhesion of young and old stem cells under conditions of increasing flow stress. Appropriately older stem cells seem less adhesive as predicted. This was related to reduced Wnt signaling in older stem cells and the adhesion of older stem cells partly restored by addition of exogenous wnt3a. The experiments seem fine. And their computational imputation of the effects of reduced adhesion shows that this might explain the clonal behaviours described earlier. The uncertainties come in the absence of any exploration of what is happening in vivo in terms of available substrates and mediators of epithelial stem cell adhesion being altered with age. Even accepting the generic interpretation, Wnt signalling is implicated in commitment and maturation processes.

Response: We provide functional data on the level of adhesion for young and aged ISCs. Our data demonstrate reduced adhesion of aged ISCs. Changes in substrates and mediators of epithelial adhesion in vivo can only imply changes in adhesion in vivo. To the best of our knowledge, it is currently not feasible to determine adhesive forces of ISCs to their environment in vivo. As the underlying nature of the reduced adhesion is likely complex (see also list of genes linked to the difference in gene expression), and modulation of adhesion in vivo that only affects 20% as predicted by the model is experimentally very challenging, the underlying mechanisms for reduced adhesion and a direct test in vivo will need to be addressed in a subsequent manuscript. This will then of course include correlative analyses on the level of substrates and mediators of adhesion.

We further agree with the reviewer that Wnt-signaling is involved many processes, including commitment and maturation, but also the regulation of adhesion. Our data demonstrate that aged ISCs obtain a more youthful adhesion upon exposure to Wnt3a, and there is precedent in the literature that Wnt-signaling affects adhesion of ISCs (Carmon *et al*, 2017). Whether

there is a mechanistic connection between adhesion and commitment and maturation induced by Wnt3a will need to await further investigations.

If stem adhesion is decreased then one might imagine that the rate of stem cell loss is increased but that this is compensated for by increased replacement rates of surviving stem cells. In which case are adhesion effects so easily divorced from expansion?

Response: Our data strongly supports this novel and central interpretation of our data by the reviewer. The model explains the accelerated drift via reduced adhesion of ISCs, while it remains fully consistent with our finding that there is no increased expansion of the initial clones upon aging.

Overall, the manuscript presents observations that are individually of interest but that do not link in a persuasive way to build a convincing story. Their initial hypothesis and final interpretation are not precisely defined. The clonal behaviours in young and old mice are not comprehensively explored. The descriptions of stem cell changes with age are conveyed qualitatively making their magnitude and their biological significance difficult to appreciate. The authors need to focus on understanding the clonal behaviours in young and old mice more fully in order to convince that adhesion effects can solely explain age related differences.

Response: We are a bit surprised by this overall summary, based on the points and comments listed before. We do not share the view that our final interpretations are not precisely defined, although we agree that some of them profited from being further focused, as now done, in this novel version of the manuscript.

We actually went to a great extent to provide quantitative data on the changes of ISC and their role in crypts, and do provide quantitative data on the magnitude on any of the observed differences between young and aged crypts and ISCs (Figures 1-4) Otherwise, we would not be able to run a mathematical model and compare it to our real-world data.

We do not imply that adhesion changes can solely explain all age-related differences, as the model only tests for the drift in the crypt, as already our title states: "contributes to" does not mean solely.

We are further surprised by the last statement of the reviewer (focus on understanding clonal behaviors), as our data follows a very different line of experiments. We do not deduct from changes in clonal behavior that changes in adhesion are an underlying cause. We use quantitative gene expression data to demonstrate differences in expression of genes linked to adhesion. We demonstrate that adhesion is indeed reduced for aged ISCs. We then use a very sophisticated, but well-established model of drift in crypts to demonstrate that indeed, reduced adhesion is able to explain very nicely the accelerated drift. As the underlying nature of the reduced adhesion is likely complex (see also list of genes linked to the difference in gene expression), and modulation of adhesion in vivo that only affects 20% as predicted by the model is experimentally very challenging, the underlying mechanisms for reduced adhesion and a direct test in vivo will need to be addressed in a subsequent manuscript.

Referee #2:

The manuscript entitled 'Reduced adhesion of aged intestinal stem cells contributes to an accelerated clonal drift' by Hageb et al. suggests a role for the strength of ISC adhesion to the niche in determining the speed of crypt fixation. The authors claim that aging results in reduced adhesion, thereby increasing the speed crypt fixation. Although this idea is highly interesting, the authors do not fully provide the experimental evidence to support their claims. In particular, the manuscript mostly confirms existing work on ISCs and aging using

the well-studied Confetti mouse model, and the novel findings on cellular adhesion are very limited. This leads to the following major concerns:

Response: We thank the reviewer on his statement that our data indeed addresses an interesting question in the field. We provide experimental evidence to support that reduced adhesion is likely a contributor to the accelerated drift. We agree with the reviewer that underlying nature of the reduced adhesion is likely complex (see also list of genes linked to the difference in gene expression), and modulation of adhesion that only affects 20% (like the model will predict and which is in part supported by our data), the underlying mechanisms for reduced adhesion and their biological outcome will need to be addressed in a subsequent manuscript.

We are though not aware of data in the literature on aging of ISCs using the confetti mouse model, or a large number of data on aging of ISCs in general (see also Nalapreddy *et al.* (Nalapreddy *et al.*, 2022)). We do further not share the statement that our findings on cellular adhesion are very limited. Our data convincingly shows reduced adhesion (novel), and the modelling convincingly shows that reduced adhesion, in the absence of changes in proliferation, is sufficient to cause an accelerated drift (novel).

1. This study demonstrates how aging increases the rate of crypt fixation in young versus old mice using the Confetti mouse model. This finding is not novel since several studies have revealed how crypt fixation is increased when the number of functional ISCs per crypt decreases, e.g. by decreasing Wnt signaling. Moreover, various studies have demonstrated that aging results in reduced ISC fitness by a decrease in Wnt signaling, which is caused by both cell intrinsic signaling and by altered secretion of niche factors, thereby generally outlining the mechanism of increased crypt fixation in aged animals. Moreover, it has been shown that addition of Wnt3a or inhibition of Wnt reducing factors rescues the effect of decreased ISC fitness, and it should therefore come as no surprise that in the current study addition of Wnt3a rescues the loss of attached cells *in vitro*.

Response: Our data is consistent with an accelerated clonal drift, which remains dynamic, which is not necessarily fixation. We feel that it is very critical to differentiate between the tumor/cancer setting and normal physiology upon aging. While we agree with the reviewer that there is information in the literature on the role of the number of ISCs for the dynamic of fixation (like for example nicely depicted also by (Huels *et al.*, 2018)), our data supports that there is no shift in the number of Lgr5+ ISCs upon aging. We actually list, to the best of our knowledge, relevant published literature on this topic in our introduction or discussion. We and others indeed demonstrate a role for a shift from canonical to non-canonical Wnt signaling for aging of ISCs. Huels *et al.* (Huels *et al.*, 2018) indeed demonstrate that systemic inhibition of Wnt ligand secretion can affect crypts dynamic, although the role of Wnt-signaling for the overall crypt architecture and dynamic remains still non fully conclusive in the field. Our data convincingly shows reduced adhesion of aged ISCs (novel), which is rescued by addition of Wnt3a (a possibility based on published data, but still novel) and the modelling convincingly shows that reduced adhesion, in the absence of changes in proliferation, is sufficient to cause an accelerated drift (novel).

2. The authors claim it is not necessarily decreased number of ISCs that causes increased crypt fixation, but a decrease in cellular adherence of ISCs. They base their conclusion on results obtained using an Lgr5-GFP mouse model, and demonstrate that the number of Lgr5-GFP cells stays equal in young versus old mice. However, as the authors pointed out in the introductory section, the number of Lgr5+ cells does not reflect the number of functional ISCs, while the number of functional stem cells dictate the rate of crypt fixation.

Response: From our finding, the number of Lgr5+-GFP ISCs are equal in young and aged crypts. Not all Lgr5+ cells from a crypt are functional, as indicated by the low organoid

formation ability of individual Lgr5+ ISCs, even when plated together with Paneth cells. As nicely shown by Huels et al. (Huels *et al.*, 2018), very low levels of Wnt signaling result in a loss of Lgr5+ cells at the border of the crypt. Very low level of Wnt signaling also result in very accelerated dynamics of drift. Both a causal and a correlative relationship between a change in number of ISCs and rate of fixation thus remain indeed likely possibilities. It might also remain a possibility that change in adhesion forces/tissue tension due to altered crypt architecture upon the loss of ISCs might contribute to change in the dynamics. Our experiments simply demonstrate accelerated drift, reduced expression of adhesion-linked genes in aged ISCs and reduced adhesion of aged ISCs on Matrigel that can be to a great extent rescued by addition of Wnt3a, and a state-of-the art ISC competition model that supports that reduced adhesion in itself affects clonal dynamics.

In addition, it remains unclear why the authors use the Lgr5 mouse model in figure 2 with the sole purpose of quantifying vertical extension, whilst this does not necessarily say something about (horizontal) ISC competition within the crypt bottom.

Response: The purpose of this experiment was indeed to focus on vertical extension and thus proliferation. In contrast to Huels et al, in our model, the induction analysis was targeted to crypts with single labeled ISCs to be able to unequivocally determine vertical extension driven by one clone, which does not allow for the analysis of inter-clone competition, which was not the purpose of the experiment.

3. The authors use scRNA-sequencing of Lgr5-GFP^{high} cells of young and old mice to observe whether there are any differences between young and old ISCs. They identify 4 distinct clusters within the Lgr5-GFP^{high} population in both young and old cells, and claim that there is increased heterogeneity within the old ISC population. However, given the fact that the same 4 clusters are identified in the young and old sample, and the complete lack of characterization of these clusters, the authors cannot make such solid claims. Moreover, it is unclear if the reduced biological adhesion as shown in figure 4d accounts for all 4 clusters in the 'old' sample, or whether a single cluster shows a lack of adhesion.

Response: We apologize if the presentation of our scRNAseq data, especially with respect to the presentation and analysis of the clusters, caused some confusion. We did not identify the same clusters in young and aged animals, only a similar number of clusters between young and aged mice. Each cluster in each animal though remains unique. The question we addressed was whether upon aging expression profiles of groups of ISCs are distinct among aged ISCs. The similar number of clusters among ISCs from individual aged and young animals demonstrates that this is not the case. We then determined the spatial distance of and the density within the clusters, which was larger among clusters of aged ISCs and denser. This is our opinion a valid and solid claim. We followed the suggestions of the reviewer and further determined the distribution of adhesion molecules in the distinct clusters to check for whether clustering is primarily driven by changes in adhesion factors.

We added this novel data to the manuscript (now SFigure5c, line 814-817). We determined the expression pattern of the GO adhesion genes (Figure 5a) in all ISCs from each individual mouse and compared it to their cluster membership (Figure 4f, SFigure 4d). The color scale shows expression, and the hierarchical clustering arranges cells according to similarities in expression levels. The lower part of the Figure identifies the cluster the cell has been associated to (young or old mice). There is no obvious pattern in the association of the genes with distinct clusters, so cells in individual clusters do not show a distinct expression pattern of adhesion genes, aka clustering is not driven by changes in adhesion molecules, but rather by differences in the expression of genes linked to overall stem cell aging (SFigure 4e).

The data supports the conclusion that there is no difference in expression levels of adhesion genes between the clusters identified in Figure 4f and SFigure4d. This type of analysis is based on ISCs from a large number of distinct crypts, it is therefore not suited to draw conclusions on behavior/adhesion of the ISCs from one crypt. Our modeling approach is also based on reduced overall adhesion of all ISCs within the model.

Figure legend for novel SFigure 5c (lines 814-817): Determination of the expression pattern of the GO adhesion genes (Figure 5a) in all ISCs from each individual mouse. The color scale shows level of expression, and the hierarchical clustering arranges cells according to similarities in expression levels of these adhesion genes. The lower part of the Figure identifies the ISC clusters of that mouse (see Figure 4f and SFigure 4d) the cell has been associated to.

4. Although the decrease in cellular adhesion as found using scRNA-seq is the most novel and interesting part of the study, the authors have not provided any data to functionally validate this finding to provide any mechanistic insight into this phenomenon. More specifically, it is unclear which adhesion molecules are reduced, whether they are reduced in some old ISCs or all ISCs, and the presence of such adhesion molecules has not been visualized either in intestinal tissue sections or in vitro organoid cultures.

Response: We thank the reviewer for the statement that our data on reduced adhesion and the modelling that is based on this data is the most interesting and novel part of the study. Our data does include the information on which adhesion-associated molecules are reduced in aged ISCs (Figure 5A). We agree with the reviewer that we are currently not able to depict what causes the reduced adhesion. The determination of changes in substrates and mediators of epithelial adhesion in vivo though can only imply changes in adhesion in vivo. To the best of our knowledge, it is currently not feasible to determine adhesive forces of ISCs to their environment in vivo. As the underlying nature of the reduced adhesion is likely complex (see also list of genes linked to the difference in gene expression), and modulation of adhesion in vivo that only affects 20% (as predicted by the modelling) is experimentally very challenging, the underlying mechanisms for reduced adhesion and a direct test in vivo will need to be addressed in a subsequent manuscript.

5. Furthermore, the authors demonstrate that old intestinal cells have reduced adhesive strength using an in vitro stress model, and that this reduction can be rescued by addition of Wnt3a. Based on these findings the authors claim that reduced adhesion is the result of decreased Wnt signaling. However, due to a lack of mechanistic insight, again the authors should not make such firm claims, as they have only demonstrated a sole correlation between Wnt signaling and adhesive strength.

Response: Our data demonstrates that if we add Wnt3a to aged ISCs (which activates canonical Wnt signaling in ISCs, see (Nalapareddy *et al*, 2017)), adhesion of aged ISCs is improved. We respectfully disagree with the reviewer that this data remains simply correlative, as providing Wnt3a to aged ISCs which show decreased Wnt signaling (Nalapareddy *et al.*, 2017) is sufficient to bring adhesion back to normal/young.

In addition, since their stress model does not reflect the architecture of the normal intestinal crypt (and probably also not the forces that are imposed on the crypt), the authors should validate their findings in an in vivo model. Moreover, to prove that cellular adhesion is a critical influencer of crypt fixation, the authors should modulate these adhesive strengths and demonstrate how this alters fixation rates.

Response: We agree with the reviewer that the extent to which our shear stress model reflects adhesive forces in vivo needs to be further verified. To the best of our knowledge, it is currently not feasible to determine adhesive forces of ISCs to their environment in vivo, so an established modelling system might be currently the best way to test the role of adhesion on clone dynamics. As the underlying nature of the reduced adhesion is likely complex (see also list of genes linked to the difference in gene expression), and modulation of adhesion in vivo that only affects 20% (as predicted by the modelling) is experimentally very challenging, the underlying mechanisms for reduced adhesion and a direct test in vivo will need to be addressed in a subsequent manuscript.

Minor concerns:

- figure 2a is not mentioned in the text

Response: We now reference Figure 2a in the text (line 169).

- figure 3 b, c, and d, are unclear in terms of figure quality and interpretation

Response: We fixed Figures 3b-d with respect to clarity/resolution, which will now allow interpretation of the data according to the description in the text.

- the GSEA plots throughout the manuscript are hard to interpret because labels of the sample are missing on the X-axis.

Response: We added labels to these X-axes.

- Figure 4h is impossible to interpret

Response: Figure 4h simply tests in as much the overall aging signature that we identified correlates with the clusters identified. We added additional information to the manuscript to further explain this to the reader. The color scale shows level of expression, and the hierarchical clustering arranges cells according to similarities in expression levels of these aging signature identified (Figure 4a-c). The lower part of the Figure identifies the ISC clusters of that mouse (see Figure 4f and SFigure 4d) the cell has been associated to. These are in total 4 clusters in mouse 9529 or 7087 (lines 732-736).

The clusters indeed correlate with the aging signature, as they form a pattern-blocks within the clusters (lower part of Figure 4f and SFigure 4d).

Referee #3:

The manuscript by Hageb and colleagues describes clonal dynamics of the intestinal stem cells in old and young mice. The authors investigate whether ISCs properties change with age using lineage tracing analyses. They find that certain stem cells are more dominant than

the others with age. Based on single-cell RNA-sequencing data, the authors conclude that cell adhesion might be important for defining the properties of aged ISCs.

Major concerns:

The 3rd sentence of the Results part: the authors use B6.Cg-Tg(Vil1-cre)1000Gum/J mice for the lineage tracing analysis either in young or old mice. The same strain is described in the methods part. They use tamoxifen to induce the expression of fluorescent reporters. However, the indicated mouse strain is not Cre-ERT but only Cre. Therefore, all conclusions in Figure 1 about the difference in clonal dynamics between the young and old mice is not relevant.

Response: We used the inducible mouse model for Vil-Cre induction, Tg(Vil-cre/ERT2)23Syr for all of the analyses in this study. We indeed listed a wrong strain in the manuscript text and also in the first section of the methods section on mice, and we need to really apologize for that mishap. The correct line is now listed in the text and in methods (lines 115 and 373).

Figure 2: how do the authors explain rather dim RFP fluorescence in this figure compared to very bright RFP in Figure 1b?

Response: In Figure 1b, all cells of the crypt are labelled and some show clonal expansion, and so frequently multiple cells with RFP are close together, which give the impression of a higher level of fluorescence than experiments shown in Figure 2, in which we analyzed crypts in which only one cells was labeled with RFP for example.

Figure 3: The sentence describing Figure 3b "Organoids formed from old crypts showed small difference in the frequency of samples in which a small number of barcodes dominated the relative abundance in old organoids" is incomprehensible.

The same is true for the conclusion sentence: "This data is consistent with no or only a very minor advantage of individual aged ISCs in conferring a dominance upon ex vivo competition that is distinct from young ISCs, even when competing against ISCs from different crypts". How could ISCs from different crypts compete ex vivo? What is this conclusion based on? I do not see the data showing competition properties for the young ISCs. The whole barcoding strategy/ outcome is not well described, and the graphs are not clear. Furthermore, ex vivo aged organoids do not have any competition advantage, which is in contradiction to the statement in the Summary.

Response: We provide now high quality graphs for the barcode analyses. We added additional text to figure and the text to better explain the barcoding experiments and the interpretation of the data stemming from these experiments.

Lines 195-211 in the manuscript: "To this end, crypts and thus ISCs in crypts from young or aged animals were transduced with the barcode library and subsequently cultured and expanded under organoid culture conditions. In these experiments, on average about 50% of organoids were transduced by the barcode virus, with a similar transduction frequency of young and aged crypts (**SFigure 3b**). The overall number of barcode libraries retrieved was similar in both age groups and showed a high level of representation of complexity (on average 75 barcodes per analysis, **SFigure 3c**). A high level of transduction and the equal level of transduction efficiency and the high level of complexity obtained upon retrieval ensured that the barcode tracing experiments are representative of all crypts in the experiment and comparable between experiments performed with young and aged crypts. Organoids formed from old crypts (O1-O5) showed no difference in the frequency of samples in which a small number of barcodes dominated the relative abundance of barcodes in a given sample (**Figure 3b**). Consistent with no significant difference in clonality parameters in young and aged organoids, the relative contribution of the most abundant barcode (**Figure 3c**), as well as the Shannon diversity index, which is a measurement of

barcode diversity (**Figure 3d**), were not different between young and aged organoids. This data is consistent with no or only a very minor advantage of individual aged ISCs in conferring a dominance upon ex vivo expansion over young ISCs.”

We agree with the reviewer that our summary statement, with respect to the outcome of the barcode experiments, was not precise enough with respect to the non-existing competition advantage of aged organoids ex vivo. The summary statement now reads “In summary, the analysis of clonal dynamics in vivo supports an accelerated clonal drift upon aging, likely due to reduced adhesion of aged ISCs due to reduced canonical Wnt signaling”, lines 342-343.

Figure 4: The authors state that GSEA analysis showed a negative correlation with ISC signature published by Munoz et al.,. I assume if they sorted and sequences stem cells the correlation must be positive.

Figure 4d and 4e do not carry much meaning.

Response: ISCs from aged animals showed indeed a negative correlation to the published stemness signature of Munoz et al. aka they are not similar to young ISCs in terms of the expression of genes associated with stemness in young ISCs. Figure 4d and 4e identify pathways enriched or de-enriched in aged ISCs compared to young ISCs. The information is critical, but we agree, the presentation of this data takes a lot of space. We though decided to keep the figure in the main figure panel and not move it to the supplementary panel, as the supplementary panel is also already quite crowded.

The gene expression differences between young and aged mice could be due to the difference in the regional identity, the anterior versus posterior, of the LGR5-EGFP ISCs. scRNA-seq UMAP or t-sne plots displaying the expression of the regional markers, such as Fabp1, Fabp2, Fabp5, Gata4, Gata6, Hmgc2, Olmf4 would be indicative that the ISCs are coming from the same regions.

Response: We agree with the reviewer that anterior versus posterior LGR5-ISCs show distinct expression profiles. As we processed whole intestine for our LGR5-ISCs, we obtained in both young and aged a random mixture of anterior or posterior ISCs, and thus also in the single cell analyses of ISCs. We did thus for example not have preferentially anterior ISCs in young and posterior ISC in aged analyses, and rather young and aged as the main difference in these analyses. This is further supported by novel data in SFigure4 that demonstrates almost identical levels of expression of regional marker suggested by the reviewer like Olmf4 among our young and aged ISCs analyzed in our analyses.

Furthermore, the expression of the certain markers for adherence junctions displayed as UMAP plots would be more convincing than GSEA plots. For example, GO-adherence junction- the enrichment score is equally high for either red (young?) and blue (old?). So, are they equal? And what are the genes included inside?

Response: We agree with the reviewer that there are distinct possibilities to present single cell expression data. With respect to a detailed analysis of change in the expression of GO genes linked to adhesion, we provide this data in Figure 5a, which to us is the most direct way to depict differential expression of genes and gene-ontologies linked to adhesion. With respect to the GSEA plot in SFigure4c, the data simply shows that adherence junction genes are downregulated in aged ISCs (see top of Figure 4c). The genes included in the GO group are listed in Figure 5a.

Figure 4F: What is Young_9529 and Old_7087? What are the 4 clusters? Why they are separated in clusters? The conclusion of Figure 4 should be strengthened by showing more data, markers. Are those few old cells in black ISCs or +4/TA cells?

Response: The numbers simply represent the identification numbers of individual animals (aka mouse 7087 in our colony). We first performed Uniform Manifold Approximation and Projection (UMAP) based dimensionality reduction followed by force-directed layout algorithm (Fruchterman and Reingold, 1991). The purpose of this analysis was thus not to link clusters to regional or sub-type specific transcription profiles of ISCs, as for example used in 10x sequencing approaches. The target was to simply determine how similar transcription profiles are among the LGR5+ISCs to determine the level of homogeneity or heterogeneity, independent of for example position information (see also above). We are confident that the conclusion from Figure 4 is valid based on the analysis of our primary data.

Figure S4F: Perhaps it is a PDF conversion, but I see that Tert and Tnfrsf19 are higher expressed in the old cells, yet the mean is lower in those cells. Could, please, the authors check for that? The same is for smoc2 and axin2, although less pronounced.

Response: We thank the reviewer for pointing this out. We very carefully re-checked presentation of the data in Figure S4F and corrected these conversion errors, which did not affect any p-value and thus statements on the significance of the difference of expression.

Figure 5a: Many genes assigned as adhesion are not encoding for adhesion molecules, such as Rgmb (L-glutamine:2-deoxy-scyllo-inosose aminotransferase), Bcl6 (transcription factor), Apc (cytoplasmic), C1qntf1 (I Beta-1,6-N-Acetylglucosaminyltransferase), etc. If aged ISCs express lower levels of integrins (or other adhesion molecules) compared to the young ISCs, it should be shown using antibody stainings on tissue sections.

Response: The reviewer is correct that not all genes that are listed within the GO group biological adhesion are direct adhesion molecules. They are though involved in the regulation of adhesiveness of cells, and thus critical part of the GO term biological adhesion. Our data demonstrates a difference in adhesion of aged ISCs to matrix, but does indeed not reveal whether it is due to differences directly in adhesion molecules on the cell surface or adhesion strength determination (mechanotransduction) in response to the adhesion signal.

We provide functional data on change in the level of adhesion for young and aged ISCs. Our data demonstrate reduced adhesion of aged ISCs. Changes in substrates and mediators of epithelial adhesion in vivo can only imply changes in adhesion in vivo. To the best of our knowledge, it is currently not feasible to determine adhesive forces of ISCs to their environment in vivo. As the underlying nature of the reduced adhesion is likely complex (see also list of genes linked to the difference in gene expression, which includes distinct types of adhesion modulators, including also integrins), and modulation of adhesion in vivo that only affects 20% is experimentally very challenging, the underlying mechanisms for reduced adhesion and a direct test in vivo will need to be addressed in a subsequent manuscript. This will then of course include correlative analyses on the level of substrates and mediators of adhesion on ISCs and the niche.

It is not clear whether the organoid assays for shearing stress were performed with CHIR or not. If with, when adding Wnt-3a (undescribed concentrations) would have no difference for the assay. As the amount of CHIR added is high enough to inhibit GSK3b to a very minimum.

Response: The shear stress experiments were performed on directly on sorted Lgr5-ISCs and not on organoids. We added the concentration of Wnt3a to Methods, which was 100ng/ml (now listed in Methods, line 483). We did use CHIR99021 (2.5 μ M) as listed in Methods (line 483). We did see a difference in adhesion when Wnt3a was added to aged ISCs. Aged ISCs show reduced canonical Wnt signaling in comparison to young ISCs, an addition of Wnt3a to

aged ISCs restores canonical Wnt signaling (Nalapareddy *et al.*, 2017), which might explain our positive response of addition of Wnt3a to aged ISCs.

The message about the involvement of adhesion in the clonal drift of ISCs is important, therefore, *in vivo* evidence, including either loss-of-function mouse models or small molecule inhibitors are desirable to confirm the computational predictions.

Response: To the best of our knowledge, it is currently not feasible to determine adhesive forces of ISCs to their environment *in vivo*. As the underlying nature of the reduced adhesion is likely complex (see also list of genes linked to the difference in gene expression, which includes distinct types of adhesion modulators, including also integrins), and modulation of adhesion *in vivo* that only affects 20% is experimentally very challenging, the underlying mechanisms for reduced adhesion and a direct test *in vivo* will need to be addressed in a subsequent manuscript.

In summary, if the adhesion is important then the dominant aged ISCs should express the adhesion molecules at higher levels compared to the excluded aged ISCs. That is not shown by the authors. There could be a difference between the young and the old, but those ISCs do not meet each other *in vivo*.

Response: We agree with the reviewer that our model on the connection between ISCs adhesion and the accelerated clonal drift implies that aged ISCs that show a reduced adhesion ability should contribute to an even faster clonal turnover. It remains though technically very challenging to determine, on the single cell level, the expression profile of excluded aged ISCs. On a population level though, we can demonstrate that aged ISCs show less adhesion. Our model clearly demonstrates that a pool of ISCs that has levels of adhesion that is lower than the control (young) group shows an accelerated drift, and the model is in very good agreement with our experimental data (Figure 1 and 2). In summary, we believe that our data indeed supports an important role for reduced adhesion for the accelerated clonal drift.

References

- Bruens L, Ellenbroek SIJ, van Rheenen J, Snippert HJ (2017) In Vivo Imaging Reveals Existence of Crypt Fission and Fusion in Adult Mouse Intestine. *Gastroenterology* 153: 674-677 e673
- Carmon KS, Gong X, Yi J, Wu L, Thomas A, Moore CM, Masuho I, Timson DJ, Martemyanov KA, Liu QJ (2017) LGR5 receptor promotes cell-cell adhesion in stem cells and colon cancer cells via the IQGAP1-Rac1 pathway. *J Biol Chem* 292: 14989-15001
- Huels DJ, Bruens L, Hodder MC, Cammareri P, Campbell AD, Ridgway RA, Gay DM, Solar-Abboud M, Faller WJ, Nixon C *et al* (2018) Wnt ligands influence tumour initiation by controlling the number of intestinal stem cells. *Nature communications* 9: 1132
- Li YQ, Roberts SA, Paulus U, Loeffler M, Potten CS (1994) The crypt cycle in mouse small intestinal epithelium. *Journal of Cell Science* 107: 3271-3279
- Lopez-Garcia C, Klein AM, Simons BD, Winton DJ (2010) Intestinal stem cell replacement follows a pattern of neutral drift. *Science* 330: 822-825
- Nalapareddy K, Nattamai KJ, Kumar RS, Karns R, Wikenheiser-Brokamp KA, Sampson LL, Mahe MM, Sundaram N, Yacyshyn MB, Yacyshyn B *et al* (2017) Canonical Wnt Signaling Ameliorates Aging of Intestinal Stem Cells. *Cell reports* 18: 2608-2621
- Nalapareddy K, Zheng Y, Geiger H (2022) Aging of intestinal stem cells. *Stem cell reports*
- Snippert HJ, van der Flier LG, Sato T, van Es JH, van den Born M, Kroon-Veenboer C, Barker N, Klein AM, van Rheenen J, Simons BD *et al* (2010) Intestinal crypt homeostasis results from neutral competition between symmetrically dividing Lgr5 stem cells. *Cell* 143: 134-144
- Winton DJ, Ponder BA (1990) Stem-cell organization in mouse small intestine. *Proc Biol Sci* 241: 13-18

April 6, 2022

RE: Life Science Alliance Manuscript #LSA-2022-01408-TR

Mr. Hartmut Geiger
University of Ulm
Department of Dermatology and Allergic Diseases, University of Ulm
James Franck Ring 11 C
Neues Forschungsgebäude, R.3010
Ulm 89081
Germany

Dear Dr. Geiger,

Thank you for submitting your revised manuscript entitled "Reduced adhesion of aged intestinal stem cells contributes to an accelerated clonal drift". We would be happy to publish your paper in Life Science Alliance pending final revisions necessary to meet our formatting guidelines.

- please provide your manuscript text in an editable doc file
- please consult our manuscript preparation guidelines <https://www.life-science-alliance.org/manuscript-prep> and make sure your manuscript sections are in the correct order
- please add an ORCID ID for both corresponding authors; you should have received instructions on how to do so
- please make sure that the author order in the manuscript and in our system match
- please double-check that all author contributions are correctly listed in our system and in the manuscript text
- please add a conflict of interest statement to your manuscript text
- please add a Data Availability Statement to include accession information for deposited datasets
- please rename your extended figures as supplementary figures in the figure legends
- please add the Twitter handle of your host institute/organization as well as your own or/and one of the authors in our system
- please upload all Video Files referred to in the manuscript text
- please add a callout for Figure S5C in your main manuscript text
- please upload your main figures as single files; these will be displayed in-line in the HTML version of your paper, so please provide them as single page files (Figure 4 currently spans 2 pages); we do not have a limit on the number of main figures and these can be split if necessary for space

A. FINAL FILES:

-- Summary blurb (enter in submission system): A short text summarizing in a single sentence the study (max. 200 characters including spaces). This text is used in conjunction with the titles of papers, hence should be informative and complementary to the title. It should describe the context and significance of the findings for a general readership; it should be written in the

present tense and refer to the work in the third person. Author names should not be mentioned.

B. MANUSCRIPT ORGANIZATION AND FORMATTING:

Sincerely,

April 12, 2022

RE: Life Science Alliance Manuscript #LSA-2022-01408-TRR

Mr. Hartmut Geiger
University of Ulm
Department of Dermatology and Allergic Diseases, University of Ulm
James Franck Ring 11 C
Neues Forschungsgebäude, R.3010
Ulm 89081
Germany

Dear Dr. Geiger,

Thank you for submitting your Research Article entitled "Reduced adhesion of aged intestinal stem cells contributes to an accelerated clonal drift". It is a pleasure to let you know that your manuscript is now accepted for publication in Life Science Alliance. Congratulations on this interesting work.

DISTRIBUTION OF MATERIALS:

Again, congratulations on a very nice paper. I hope you found the review process to be constructive and are pleased with how the manuscript was handled editorially. We look forward to future exciting submissions from your lab.

Sincerely,
